# Dehydration-driven stress transfer triggers intermediate-depth earthquakes

Thomas P. Ferrand[1], Nadège Hilairet[2], Sarah Incel[1], Damien Deldicque[1], Loïc Labrousse[3], Julien Gasc[1], Joerg Renner[4], Yanbin Wang[5], Harry W. Green II[6] & Alexandre Schubnel[1]

Intermediate-depth earthquakes (30–300 km) have been extensively documented within subducting oceanic slabs, but their mechanics remains enigmatic. Here we decipher the mechanism of these earthquakes by performing deformation experiments on dehydrating serpentinized peridotites (synthetic antigorite-olivine aggregates, minerals representative of subduction zones lithologies) at upper mantle conditions. At a pressure of 1.1 gigapascals, dehydration of deforming samples containing only 5 vol% of antigorite suffices to trigger acoustic emissions, a laboratory-scale analogue of earthquakes. At 3.5 gigapascals, acoustic emissions are recorded from samples with up to 50 vol% of antigorite. Experimentally produced faults, observed post-mortem, are sealed by fluid-bearing micro-pseudotachylytes. Microstructural observations demonstrate that antigorite dehydration triggered dynamic shear failure of the olivine load-bearing network. These laboratory analogues of intermediate-depth earthquakes demonstrate that little dehydration is required to trigger embrittlement. We propose an alternative model to dehydration-embrittlement in which dehydration-driven stress transfer, rather than fluid overpressure, causes embrittlement.

[1] Laboratoire de Géologie, CNRS UMR 8538, Ecole Normale Supérieure, PSL Research University, 75005 Paris, France. [2] Unité Matériaux et Transformations - UMR 8207, CNRS, Univ. Lille, ENSCL, INRA, F-59000 Lille, France. [3] Institut des Sciences de la Terre de Paris, Université Pierre et Marie Curie, 75005 Paris, France. [4] Institut für Geologie, Mineralogie und Geophysik, Ruhr Universität Bochum, Bochum D44780, Germany. [5] Center for Advanced Radiation Sources, University of Chicago, Argonne, Illinois 60439, USA. [6] Department of Earth Science, University of California, Riverside, California 92521, USA. Correspondence and requests for materials should be addressed to T.P.F. (email: ferrand@geologie.ens.fr).

Intermediate-depth earthquakes (30–300 km) occur mostly at subduction zones in the so-called upper and lower Wadati-Benioff planes of seismicity (UBP and LBP)[1–3]. The LBP is located in the mantle of the subducted oceanic lithosphere[4,5], 20–40 km below the plate interface. Several mechanisms have been proposed to explain the mechanics of these earthquakes, which take place under pressure and temperature conditions at which rocks are supposed to yield plastically rather than to behave in a brittle manner: (1) dehydration-embrittlement of antigorite[2,4–6], the main hydrous mineral present in the subducted plate and thus one of the most important water carriers in a subduction-zone environment, (2) quasi-adiabatic shear-heating instabilities[7,8], (3) reactivation of pre-existing shear zones[9] or (4) a combination of these.

Seismic imaging has generally documented the absence of substantially reduced velocities deep below the oceanic Moho, thus demonstrating little hydration if any[9]. However, recent seismic surveys revealed deep reflections interpreted as bending-related faulting and mantle serpentinization at the Middle America trench[10] and offshore Alaska[11]. These observations document that the lithospheric mantle was partially hydrated 8 and 15 km below the Moho, respectively, through serpentinization of deep faults, which can be extended to 35 km depth based on an estimate of the brittle-ductile transition of the lithosphere[10]. Evidence of the direct link between mantle hydration and the generation of dehydration-induced intermediate-depth seismicity is also demonstrated[11] offshore Alaska.

Antigorite, the high-temperature serpentine variety, is the most abundant hydrous mineral in the mantle in a subduction-zone environment. The stability field of antigorite is thus commonly used to predict the depth where fluid is released in subduction zones[4,5].

At the laboratory scale, experiments performed on serpentinites up to 6 gigapascals (GPa) confining pressure, relying on acoustic emissions (AEs) as a proxy for dynamic shear fracture propagation, showed contrasting results regarding dehydration-induced seismicity. Some authors concluded that antigorite dehydration is potentially seismogenic, even at the laboratory scale[12–14]. More recent studies challenged the dehydration-embrittlement hypothesis, reporting that antigorite dehydration proceeding under stress produces distributed deformation[15,16] or stable fault slip[15,17,18] without acoustic activity[15,19].

Here, we dehydrated synthetic antigorite-olivine aggregates during deformation at upper mantle conditions while monitoring AEs. Deformed specimens were investigated post-mortem using scanning and transmission electron microscopy. Focusing on the degree of serpentinization (antigorite content), we propose a new view on the failure mechanism at high-pressure. A conceptual model, where embrittlement arises from dehydration-driven stress transfer rather than fluid overpressure, harmonizes our understanding of the rheology of partially serpentinized mantle under subduction conditions. We propose this chemo-physical process as a mechanism for the LBP seismicity.

## Results

**Experimental plan and conditions.** Synthetic serpentinized peridotites were hot-pressed from mixtures of sieved ($<30\,\mu m$) olivine and antigorite powders at about 1.5 GPa and 773 K in a piston-cylinder apparatus for 10 h with antigorite fractions of 0, 5, 20 and 50 vol%. We consider these fully dense synthetic aggregates (Fig. 1a) as proxies for partially serpentinized peridotites. Cylindrical specimens of 2.1 mm diameter and 3 mm length were deformed in a D-DIA[20] equipped with an acoustic emission

detection system[21] (cell assembly in Fig. 1b). Stress and reaction progress were measured in situ using X-ray diffraction (Fig. 1c,d). Strain was measured using radiography. Six sensors continuously monitored AEs[15,21].

Deformation conditions were chosen near the stability limit of antigorite and representative of pressure and temperature conditions at the hypocenters of intermediate-depth seismicity ($1 < P < 3.5\,GPa$, $773 < T < 1073\,K$). In total, eight deformation experiments were conducted (Table 1). One set of samples was pressurized to 1.1 GPa and the second one to 3.5 GPa to explore the possible mechanical effect of volume expansion and contraction, respectively, associated with antigorite dehydration (which produces fluid plus solid), as the Clapeyron slope of the dehydration reaction changes from positive to negative over this pressure range (Fig. 2).

Samples were deformed, first, at a constant temperature of $\sim 773\,K$. After 10% strain, temperature was increased at a rate $\dot{T} \sim 4\,K\,min^{-1}$ to induce syndeformational dehydration. In one experiment, heating started at 4% strain (Table 1). The strain rate amounted to $\dot{\varepsilon} \sim 5 \times 10^{-5}\,s^{-1}$ (axial shortening) all through. The ratio of heating rate to strain rate, $\dot{T}/\dot{\varepsilon}$, of about 1,250 K, is within the range of values calculated for real subducting slabs ($100\text{–}10,000\,K$)[17].

The maximum differential stress at 773 K decreases with increasing antigorite volume fraction, in good agreement with experimental studies performed at lower pressure than applied here showing that serpentine presence strongly impacts the rheology of altered peridotites[22]. Softening is systematically observed during heating (Fig. 3).

Acoustic emissions (AEs) occurred from samples with antigorite contents, as low as 5 vol% and with up to 50 vol%, deformed at pressures of 1.1 and 3.5 GPa, respectively (Figs 2 and 3). These AEs, occurring once the reaction line was crossed, reveal that dynamic shear fractures nucleated and propagated during antigorite dehydration. In pure olivine, a few AEs were recorded at 3.5 GPa only, at the maximum differential stress, that is, at the beginning of temperature ramping (Fig. 3). The number of AEs detected in this pure olivine sample is not significantly different (although lower) than in the case of the antigorite-bearing samples. However, AEs were detected at a much higher stress level than for the antigorite-bearing samples and actually at stress levels higher than expected to occur in a cold subducting lithosphere. In addition, our synthetic samples were sintered at relatively low P conditions (1.5 GPa) so that stress concentrations may exist and play a role in the AE triggering at 3.5 GPa.

**Antigorite stability and thermal breakdown.** The characteristics of antigorite dehydration reactions (such as temperature and Clapeyron slope) vary with pressure. We recall here the main features of the phase diagram so as to compare with our experimental results.

The exact antigorite chemical formula is $M_{(3m-3)}T_{(2m)}O_{5m}(OH)_{(4m-6)}$ where $m = 14\text{–}23$, $M$ and $T$ are octahedral and tetrahedral sites. For the sake of simplicity, here we use the approximate formula $Mg_3Si_2O_5(OH)_4$. The complete antigorite (atg) dehydration reaction into olivine (fo, forsterite = Mg-endmember), enstatite (en) and water, with approximate stoichiometry, is:

$$Mg_3Si_2O_5(OH)_4(Atg) \rightarrow Mg_2SiO_4(Fo) + MgSiO_3(En) + 2H_2O(fluid) \tag{1}$$

Below 1 GPa, experiments[23] have shown that antigorite dehydration is a two-step process (Fig. 2), which first produces talc, at temperatures ranging between 873 and 973 K depending

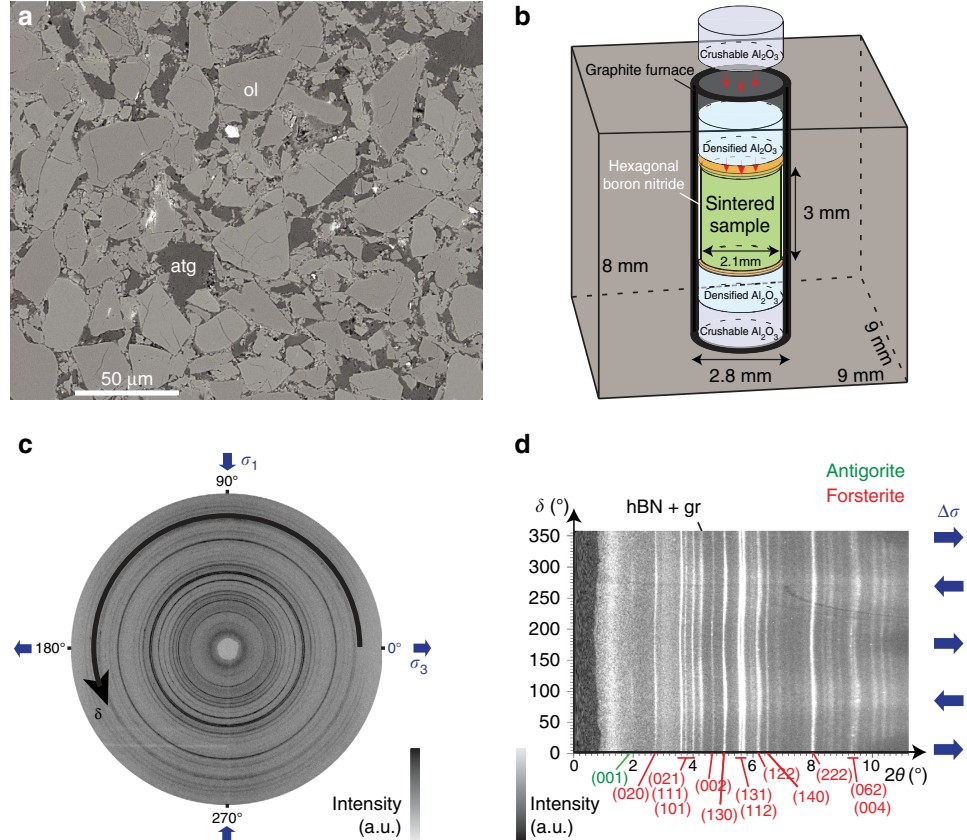

**Figure 1 | Experimental set-up. (a)** SEM micrograph of a sintered sample showing homogeneous microtexture and grains distribution: $\sim 80$ vol% SC olivine (ol) $+ \sim 20$ vol% Corsica antigorite (atg). **(b)** D-DIA assembly in a boron-epoxy cube. The sintered sample and its hexagonal boron nitride sleeve are inserted in the graphite furnace, in between alumina pistons. Boron-epoxy cube and hBN are transparent to X-rays. Gold foils (yellow) are used as strain markers. **(c)** X-ray diffraction pattern used to extract the stress from the lattice strains. For a sample with only 5 vol% of antigorite and 95 vol% of olivine. **(d)** Unwrapped diffraction pattern and indexing of diffraction peaks.

**Table 1 | Summary of experiments.**

| Exp. | Sample | $\phi_{atg}$ (%) | $\sigma_m$ (GPa) | $\Delta\sigma_{max}$ (GPa) | $\Delta\sigma_{max}/\sigma_3$ | AEs nb. | $\varepsilon$ at 923 K (%) |
|---|---|---|---|---|---|---|---|
| D1665 | 00-1.2 | 0 | $1.2 \pm 0.1$ | $2.2 \pm 0.1$ | $1.8 \pm 0.1$ | 1 | 21 |
| D1617 | 05-1.1 | 5 | $1.1 \pm 0.1$ | $1.5 \pm 0.1$ | $1.4 \pm 0.1$ | 37 | 22 |
| D1619 | 20-1.1 | 20 | $1.1 \pm 0.1$ | $0.9 \pm 0.1$ | $0.8 \pm 0.1$ | 0 | 14 |
| D1777 | 20-1.2 | 20 | $1.2 \pm 0.1$ | $1.4 \pm 0.1$ | $1.2 \pm 0.1$ | 0 | 21 |
| D1778 | 00-3.5 | 0 | $3.5 \pm 0.1$ | $2.8 \pm 0.1$ | $0.8 \pm 0.1$ | 8 | 23 |
| D1659 | 05-3.5 | 5 | $3.5 \pm 0.1$ | $1.9 \pm 0.1$ | $0.5 \pm 0.1$ | 12 | 21 |
| D1662 | 20-3.5 | 20 | $3.5 \pm 0.1$ | $2.0 \pm 0.1$ | $0.6 \pm 0.1$ | 16 | 22 |
| D1779 | 50-3.5 | 50 | $3.5 \pm 0.1$ | $1.3 \pm 0.1$ | $0.4 \pm 0.1$ | 14 | 21 |

Conditions and sample characteristics for the eight experiments performed under Synchrotron X-ray beam: antigorite fraction $\phi_{atg}$, initial mean stress at the beginning of deformation $\sigma_m$, differential stress $\Delta\sigma_{max}$, ratio between effective and differential stresses, number of recorded AEs during the experiment and strain $\varepsilon$ at 923 K. The first column shows the name of the experiment in GSECARS experimental database. The second column shows the sample name, defined with antigorite content and confining pressure.

on the pressure, in the following way:

$$5\text{Atg} \rightarrow 6\text{Fo} + \text{Mg}_3\text{Si}_4\text{O}_{10}(\text{OH})_2(\text{Talc}) + 9\text{H}_2\text{O}(\text{fluid}) \quad (2)$$

At even higher temperatures, talc reacts with olivine to produce enstatite, which releases additional water:

$$6\text{Fo} + \text{Talc} \rightarrow 5\text{Fo} + 5\text{En} + \text{H}_2\text{O}(\text{fluid}) \quad (3)$$

At pressures greater than 2 GPa, complete dehydration reaction was experimentally observed to occur in two steps too, now involving an intermediate phase, generally referred to as

'talc-like'[23,24] (Fig. 2):

at low temperature$(800 - 850\,\text{K})$ :
$$\text{Atg} = \text{Fo} + \text{'talc like'} + \text{H}_2\text{O} \quad (4)$$

at high temperature$(900 - 950\text{K})$ :
$$\text{Fo} + \text{'talc like'} = \text{En} + \text{Fo} + \text{H}_2\text{O} \quad (5)$$

The two sets of reactions exhibit one fundamental difference: reaction (2) has a positive Clapeyron slope and is endothermic, producing a net total (solid + fluid) volume increase between the initial and final reaction products; the Clapeyron slope of reaction (3) is negative, and the reaction is associated with a net total

(solid + fluid) volume decrease between the initial and final reaction products. Above 2 GPa, both reactions of the sequence (4) and (5) exhibit negative Clapeyron slopes, and thus negative volume changes.

In our experiments, antigorite dehydration was observed between 873 and 973 K at 1.1 GPa and between 873 and 923 K at 3.5 GPa (Fig. 2). Onsets of reactions (2) ('tl-in') and (3) ('Atg –', see the legend in Fig. 2), were observed at a slightly lower temperature than in kinetics studies[23,24]. Antigorite also contains traces of ferric iron ($Fe^{3+}$), so that iron oxides $Fe_2O_3$ are also often produced during the dehydration reaction. Iron decreases reaction temperatures. Some octaedral and tetraedral sites can also be occupied by aluminium ions ($Al^{3+}$) in antigorite, possibly increasing reaction temperatures[25]. Moreover, due to our temperature calibration method on axially undeformed cells assemblies, temperature might be overestimated during deformation. Indeed, sample shortening along the compression axis reduces the distance between the top and bottom WC anvils, which are efficient heat sinks[26].

On the contrary, reaction products were detected by X-ray diffraction at temperatures higher than those determined by kinetics studies[23,24]: a talc-like phase appeared between 723 and 923 K irrespective of pressure, and enstatite was only detected at temperatures higher than 953 K at 1.1 GPa (Fig. 2). These oversteps in temperature can likely be attributed to the fast heating relative to slow reaction kinetics, in particular that of grain nucleation/growth. Finally, a fast decrease of the antigorite diffraction peaks (Fig. 2, 'Atg –') indicates an acceleration of antigorite dehydration, thus promoting the completion of reaction (2) or (4) and the production of talc or 'talc-like' phase.

**Acoustic activity during experiments.** Occurrence of AE signals just above the dehydration temperature is a characteristic feature of our experiments (Fig. 2). At 1.1 GPa, AEs are observed contemporaneously with the growth of enstatite, that is, reaction (3). At 3.5 GPa, AEs are observed from the appearance to the disappearance of the talc-like phase, that is, continuously during the dehydration process. We observed AEs during dehydration at 1.1 GPa only in the sample with 5 vol% antigorite. At 3.5 GPa, however, AEs were recorded in samples with antigorite fractions from 5 to 50 vol% (Figs 2 and 3c,d).

In total, 79 AEs were recorded (Table 1) in dehydrating antigorite-olivine aggregates. The frequency of relative moment magnitudes ($M_{AE}$) follows a Gutenberg–Richter distribution ($\log_{10} N = a - b M_{AE}$) (Fig. 4a) with $b \sim 0.6$. Fast Fourier transform of the waveforms reveals a corner frequency $f_c \sim 2$ MHz for the largest AEs (Fig. 4b). Assuming rupture velocities in olivine close to shear wave velocity $v_r \sim 5$ km s$^{-1}$, the corresponding fault length ($L \sim v_r/f_c$) is $\sim 2.5$ mm, that is, of the order of the specimen size. Here, only the largest faults going through the whole sample (Figs 5a–c and 6) have measurable (low enough)

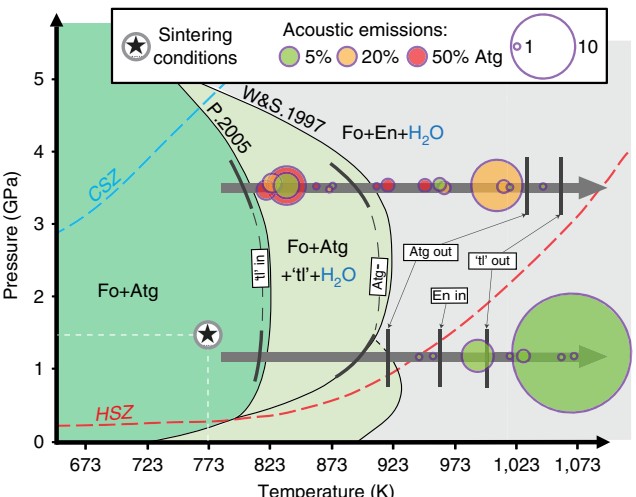

**Figure 2 | Phase diagram and AE triggering.** Stability field of antigorite in the MSH system, approximate experimental P–T path (grey arrows) and acoustic-emission recording (circles). CHZ/HSZ: geotherms on top of the slab in Cold/Hot Subduction Zone. Fo, forsterite; Atg, antigorite; En, enstatite; 'tl', 'talc-like' phases. P.2005 (ref. 23): 'talc-like' 'in' (in reaction (4)); W&S.1997 (ref. 47): enstatite in. Thick black lines correspond to initiation and termination of reactions observed experimentally, 'Atg –' to the fast destabilization of antigorite corresponding to the beginning of reaction (5).

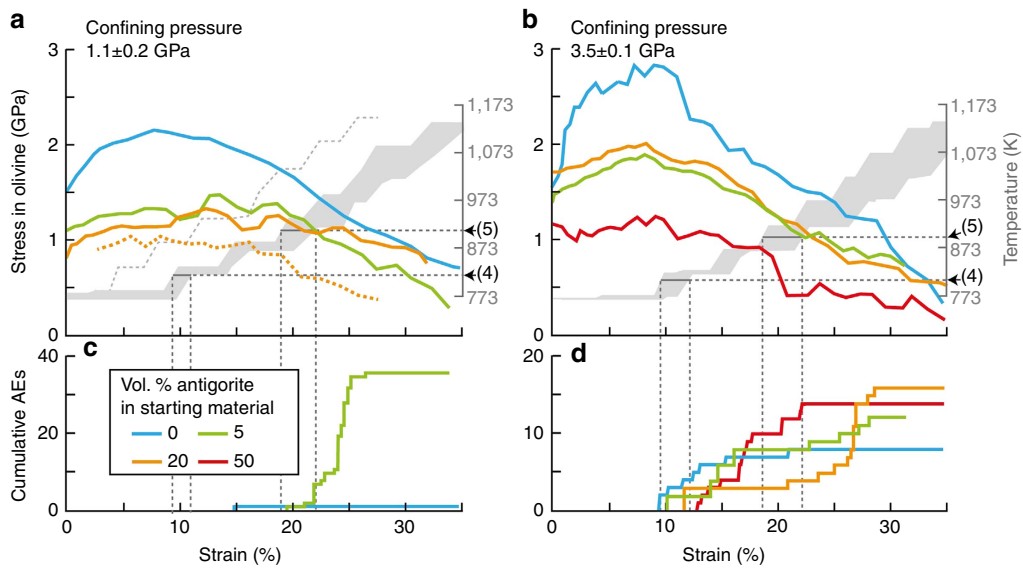

**Figure 3 | Stress evolution and acoustic activity as a function of strain and temperature.** Average differential stress in olivine as a function of strain, calculated from synchrotron X-ray diffraction patterns, for experiments at 1.1 ± 0.2 GPa (**a**) and 3.5 ± 0.1 GPa (**b**) along with temperature evolution. Associated cumulative acoustic emissions are shown in **c**,**d**. Temperature limits for onsets of reactions (4) (ref.23) and (5) (ref. 47) respectively relate to antigorite destabilization.

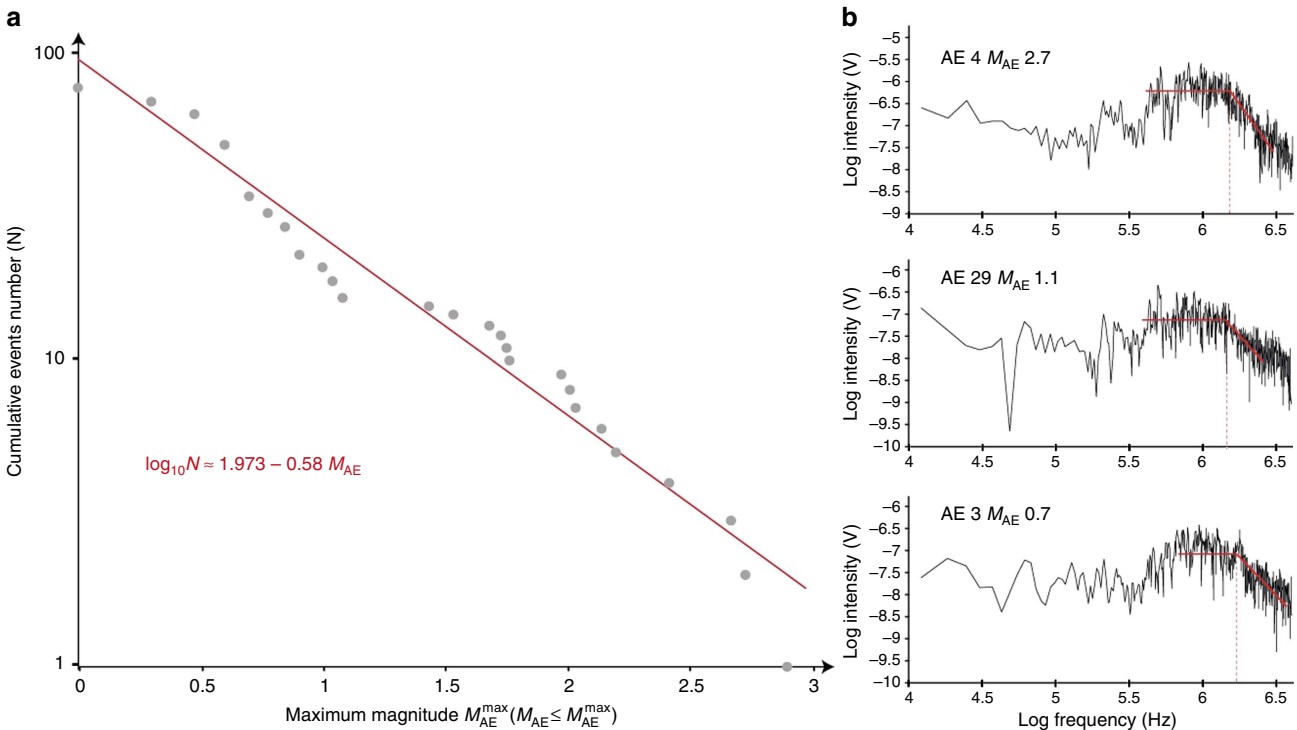

**Figure 4 | Acoustic activity during antigorite destabilization. (a)** Statistical distribution of the relative magnitudes $M_{AE}$ for the entire AE catalogue recorded. The magnitude is calculated as $M_{AE} = \log_{10}E$ where $E$ is the surface of the recorded waveform ($V^2$), considered proportional to the radiated acoustic energy. (**b**) Log–log Fourier transforms of three AEs recorded on sensor 3, behind a tungsten carbide anvil. Signals display a corner frequency $f_c$ of about 2.5 MHz on average, irrespective of the AE magnitude.

corner frequencies, because $f_c$ is close to the limit frequency of our transducers. The maximum AE magnitude size in our sample indeed corresponds closely to the size of our specimen as outlined in microstructural observations section below. The minimum value of the AE magnitude we detected probably corresponds to our detection limit. The Gutenberg–Richter scaling thus suggests that larger AE magnitudes would have been observed if our sample were larger. It has been recently shown that the fracture energy dissipated during an earthquake rupture scales with the size of the seismic asperities[27], with an apparent continuous scaling law from the laboratory scale, to the field scale. Thus, our observation of three orders of magnitude of seismic source sizes (Fig. 4a) suggests that our observations can be upscaled.

**Microstructural observations**. Scanning electron microscopy (SEM) images, taken in back scattered electron mode, revealed a population of thin faults for antigorite-bearing samples that produced AEs (Fig. 5a–f). Samples with 20 vol% antigorite deformed at 1.2 GPa (Fig. 5g–h) exhibit homogeneous deformation, with intensively sheared antigorite pseudomorphs, probably markers of strain localization. The pure-olivine sample deformed at 3.5 GPa (Fig. 5i–k) shows an intriguing microstructure, with shear bands, about 50–100 µm thick, crosscutting the sample. These shear bands are filled with angular sub-micrometric to micrometric olivine grains. However, in this antigorite-free sample, AEs were less numerous, detected at the highest macroscopic deviatoric stress only, which corresponded to the beginning of temperature ramping (Fig. 3). Large cataclastic shear bands might point towards a possible role played by thermal stress relaxation in olivine crystals (or the assembly) under high stress. Wide cataclastic shear bands are rarely associated with a single dynamic (rapid) shear failure, because the elastic energy is converted into surface energy and dissipated through fractures in the shear zone rather than radiated at high speed. It is also

possible that further heating and deformation overprinted the actual microstructure on failure. We did not further investigate the shear bands observed in this single pure olivine sample.

In comparison to faults in the pure olivine sample, faults observed in antigorite-olivine samples that produced AEs during dehydration are thinner and filled with a sub-micrometric material, which cannot be resolved at the SEM scale (Figs 5a–f and 7). We conclude that a different failure mechanism operated at lower stress in the dehydrating samples than in the pure olivine sample. Individual faults are ∼1 mm long, with sealed slip surfaces (Fig. 7b,c) typical of high-pressure faults[21,28]. Displacement markers indicate a minimum of 10 µm offset (Figs 5a–f, 6 and 7b). Using $M_0 = \mu DS$ (ref. 29) with $\mu \sim 80$ GPa (shear modulus of olivine), $D \approx 10$ µm (total slip), and $S \sim 10^{-6}$ m$^2$ (fault surface area), seismic moment is estimated to be $M_0 \approx 0.8$ Nm and the associated magnitude $M_w \approx -6$. The associated stress drop (for a circular crack expanding at constant velocity) $\Delta\sigma = C\mu D/L$, with fault length $L = 1$ mm and a geometric constant $C \sim 1$, results to about 800 MPa, that is, an almost complete stress drop, well above the maximum values inferred by seismological studies of intermediate-depth earthquakes up to date[30]. Such a large stress drop is also above the resolution of the X-ray diffraction stress measurement (<100 MPa) and could go unnoticed if either the fault did not develop on the path of X-rays or did not cross-cut the entire specimen, which is the case of the particular example we show (Figs 5a–c, 6 and 7).

Transmission electron microscopy (TEM) images, taken in bright field mode, highlight details of the fault zone observed in the sample that initially contained 5 vol% of antigorite and deformed at 1.1 GPa (Fig. 8). Chemical analyses (using SEM in back scattered electron mode, Fig. 7d–h) shows that the fault (Fig. 6) is filled by a material slightly depleted in magnesium (Fig. 7f), which is a strong evidence for antigorite dehydration

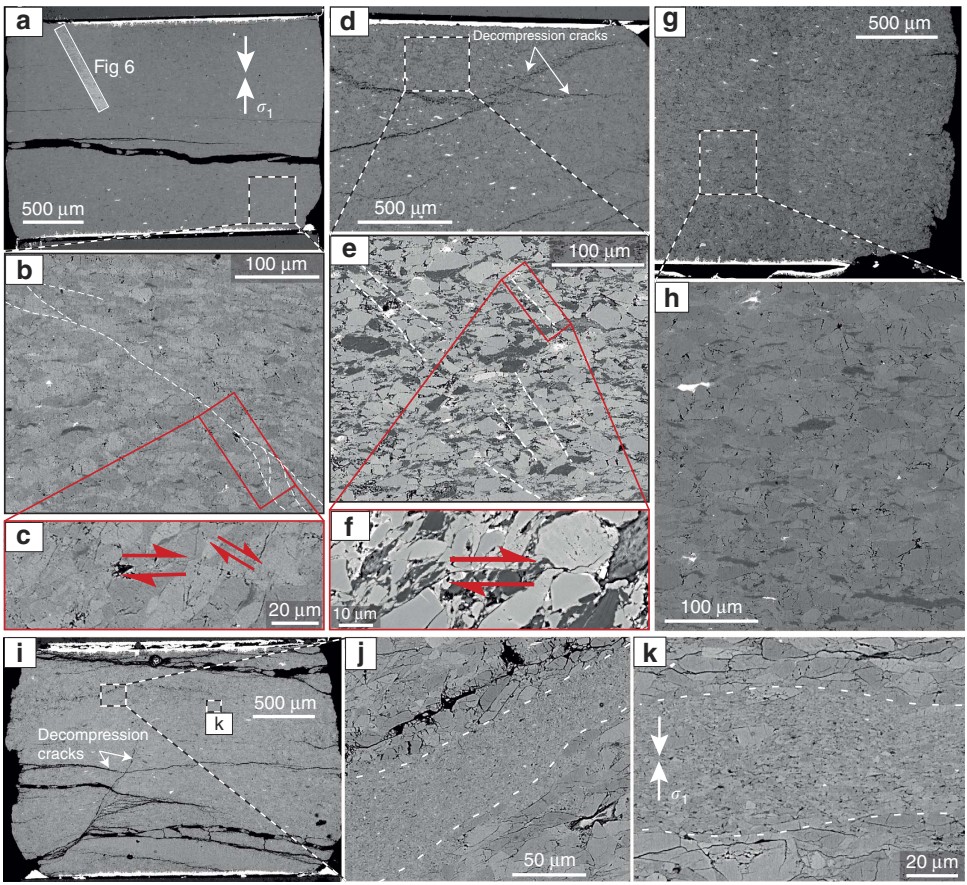

**Figure 5 | SEM micrographs of samples.** (**a**–**c**) 05–1.1, (**d**–**f**) 50–3.5, (**g**,**h**) 20–1.2 and (**i**–**k**) 00–3.5. (**a**) Sample 05–1.1 with location of Fig. 6; (**b**) sealed fault ; (**c**) intensely sheared grains along a fault; (**d**) sample 00–3.5; (**e**) faults; (**f**) microfault in olivine and antigorite between two antigorite pseudomorphs; (**g**) sample 20–1.2; (**h**) detail showing homogeneous texture of sample 20–1.2 with deformed (flattened) antigorite pseudomorphs; (**i**) sample 00–3.5; (**j**) shear zone of intensely reduced grain size; (**k**) detail of the shear zone, exhibiting angular micrometric olivine grains. $\sigma_1$ refers to the maximum compression axis.

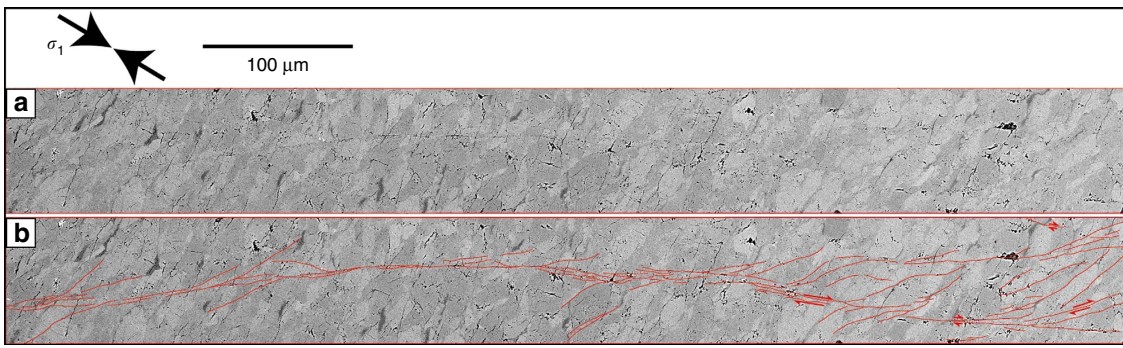

**Figure 6 | SEM micrograph of the HP-fault.** Fault is located in Fig. 5a and detailed in Figs 7 and 8. (**a**) Plain image; (**b**) interpreted fault structure indicated with red lines.

products, such as talc or talc-like phases[23]. The TEM images of focused-ion beam foils (Fig. 8) provides a closer view on the dehydration products, especially iron-oxide spherulites and fluid vesicles (Fig. 8a–c). We assume that the light phase corresponds to the 'talc-like' phase, known to be highly disordered[23]. A 20–200 nm thick gouge, composed of amorphous material, nanometric crystals and vesicles containing fluid (Fig. 8d–g) is filling the fault itself. Fluid vesicles look similar to the ones recently observed during flash melting at asperity contacts in stick-slip experiments on pure antigorite at lower pressure than

applied here[31]. Finally, in a 1 µm-thick fracture connected to the fault (Figs 7c and 8h–j), euhedral olivine nanocrystals exhibit triple junctions wetted by an amorphous phase resembling microstructures of quenched partially molten peridotites[32].

## Discussion

Altogether, the microstructural observations strongly suggest that the fault surface melted during the experiment[31,33] and, therefore, the fault could be referred to as a micro-pseudotachylyte. In turn,

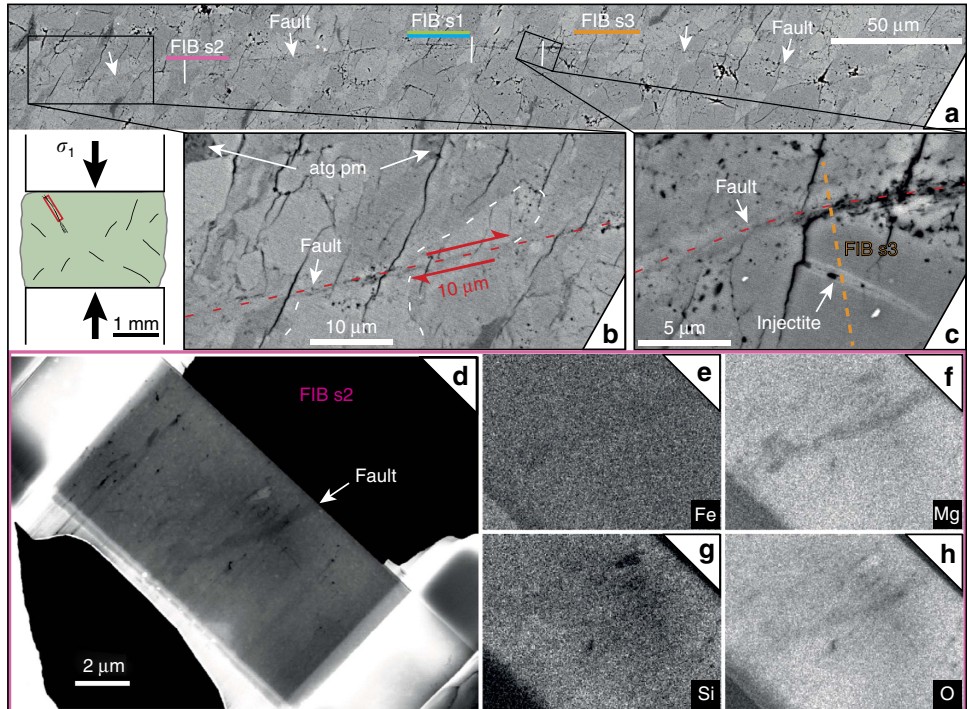

**Figure 7 | Microstructural observations in a sample with 5 vol% antigorite deformed at 1.1 GPa.** (**a**) Fault trace (complete in Fig. 6) with locations of FIB sections; (**b,c**) zooms into the fault: antigorite pseudomorphs (atg pm) and sheared grain (displacement ∼10 μm (**b**); position of FIB s3 (Fig. 8), crosscutting an injectite (**c**); (**d–h**) SEM on FIB section 2: density contrast (**d**) and EDX mapping (Fe, Mg, Si, O), showing that the fault is depleted in magnesium (**e–h**).

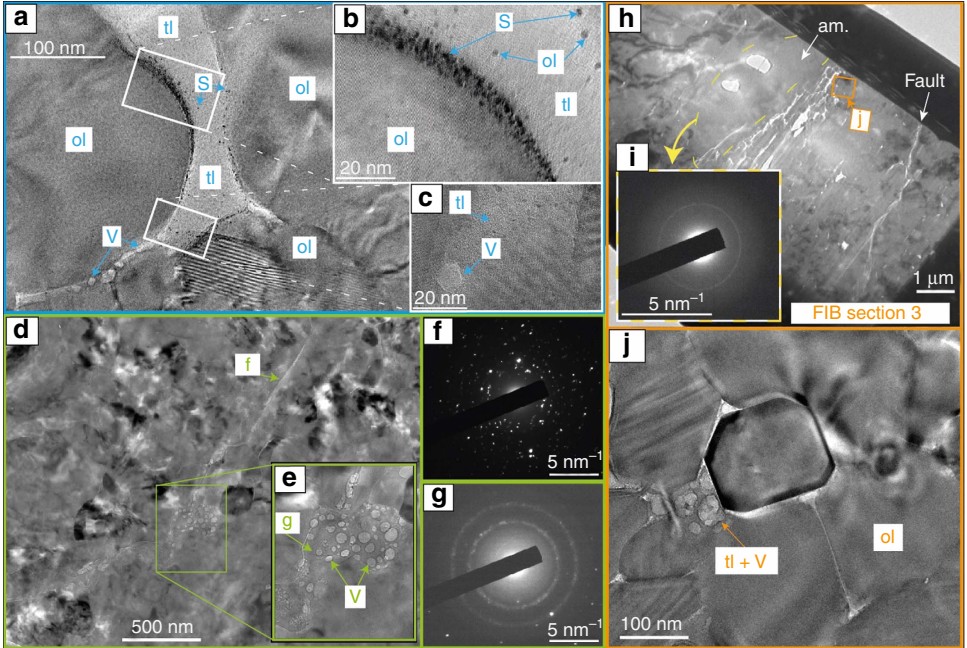

**Figure 8 | TEM images taken in bright field mode.** Sample 0.5–1.1. (**a**) Away from the fault (>1 μm), TEM image showing iron oxides spherulites (S) and disordered weak phase (tl), as products of antigorite dehydration, along with light vesicles (**b**) detail showing olivine and iron oxides crystals; (**c**) vesicle within disordered phase in between two olivine grains. (**d**) FIB section 1; (**e**) TEM image showing the fault contains vesicles (V), which decrepitate under TEM beam (200 keV) and likely contain dehydration fluids; (**f**) X-ray diffraction pattern showing nanocrystalline microstructure near the fault interface; (**g**) X-ray diffraction pattern showing amorphous material surrounding fluid inclusions. (**h**) TEM image of the FIB section 3; (**i**) X-ray diffraction pattern in the injectite, showing amorphous material over a surface of ∼10 μm², that is, significantly larger than the fault; (**j**) euhedral olivine crystals and wetted triple junctions on a side of the injectite, near the fault (holes due to ion beam).

the occurrence of pseudotachylyte in our millimetric samples is evidence for both fast shear rupture propagation and complete fault lubrication[34]. Assuming that all of the mechanical work $W$ is converted into heat $Q$, the maximum thickness of the molten zone follows the relation: $w = \bar{\tau} \times D / \left[ \rho \times \left( \Delta T \times c_{\mathrm{p}} + H \right) \right]$ (ref. 34), where $\bar{\tau} \sim 500\,\mathrm{MPa}$ (average shear stress, that is, differential stress at which slip proceeds), $\rho \sim 3.3 \times 10^{3}\,\mathrm{kg\,m}^{-3}$ (density), $\Delta T \approx 1{,}000\,\mathrm{K}$ (temperature difference to melting point), $c_{\mathrm{p}} \approx 1\,\mathrm{kJ\,kg}^{-1}\,\mathrm{K}^{-1}$ (heat capacity), $H \approx 0.3\,\mathrm{MJ\,kg}^{-1}$ (latent heat of fusion). A coseismic displacement $D \approx 2\,\mu\mathrm{m}$, that is, about only 20% of the total slip deduced from strain markers, suffices to produce a melt layer of $w \sim 200\,\mathrm{nm}$, that is, the width of observed 'fault gouge'. This could point to a high-seismic efficiency $\chi$ (here defined as $\chi = 1 - Q / W \approx 0.9$), suggesting that about 90% of $W$ was transformed into fracture energy $E_{\mathrm{G}}$ or radiated energy $E_{\mathrm{R}}$ through elastic waves. Nevertheless, except for a few micrometric injectites, little fracturing could be specifically attributed to high-pressure and temperature faulting in our specimen. Furthermore, because mode I cracking is prevented at such high normal stress, we expect the fracture energy $E_{\mathrm{G}}$ to be small compared to $E_{\mathrm{R}}$. In turn, this suggests that the seismological radiation efficiency $\eta = E_{\mathrm{R}} / (E_{\mathrm{G}} + E_{\mathrm{R}})$ would be greater here than the one usually observed by seismologists for intermediate depth seismicity[30].

However, the efficiency discrepancy and apparent paradox of the absence of large stress drop in our stress measurements can be easily reconciled in the two following manners: (1) only 10–20% of the total observed fault slip was produced co-seismically, and most of the observed slip occurred as aseismic after-slip; or (2) complete stress drops are transient dynamic phenomena, immediately followed by quenching-driven re-strengthening. Indeed, if complete stress drops occurred over a fraction of a (milli)second at the (laboratory) earthquake rupture scale, and if they are followed by immediate re-strengthening, they would go unnoticed both at laboratory scale because of our X-ray diffraction detection limits, and at the field scale because of the current seismological techniques used to invert for stress drops. In fact, earthquakes stress drops are generally inverted from estimates of the corner frequency and thus correspond to an apparent average final—static—stress drop accommodated over the entire rupture plane[30]. Recent laboratory measurements of dynamic stress drops[31] and high-velocity frictional behaviour in serpentine[33] suggest that this second solution is most likely. For a radiation efficiency $\eta \approx 0.3$ observed for intermediate-depth earthquakes[30], the average shear stress $\bar{\tau} \approx \left[ \rho \times \left( \Delta T \times c_{\mathrm{p}} + H \right) \right] / \left[ (1 - \eta) D \right] \times w$ during sliding is below $100\,\mathrm{MPa}$, which confirms that most of the slip occurred by complete fault lubrication.

The typical dehydration-embrittlement model considers the reduction in effective stress associated with the increase in fluid pressure as the main factor for shear instability. The conditions the most propitious to this fluid-driven embrittlement are (1) a positive reaction volume change (that is, $P < 2\,\mathrm{GPa}$) increasing the fluid pressure and (2) a large amount of antigorite. Here, AEs were observed below $2\,\mathrm{GPa}$ only with low-antigorite fraction (5 vol%), and only after reaction (5) and at conditions of talc breakdown, that is, a reaction associated with a negative volume change (Fig. 2). Antigorite is observed up to $\approx 920\,\mathrm{K}$. Above $2\,\mathrm{GPa}$, the total (solid and fluid) volume change associated with both steps of the antigorite-dehydration reaction is negative, so fluid overpressure is not expected, except if large amounts of compaction occur. At $3.5\,\mathrm{GPa}$, AEs were nevertheless triggered on dehydration, continuously during both steps of the reaction. Systematic dehydration-induced AE triggering was also observed independently of antigorite fraction (5–50 vol %). At $1.1\,\mathrm{GPa}$, the $\Delta V$ of the reaction is positive ($\approx +5\%$) (ref. 15), which tends to prevent tensile stress accumulations in the olivine skeleton; at

$3.5\,\mathrm{GPa}$, the $\Delta V$ is negative ($\approx -1\%$)[15], which promotes tensile stress accumulation at cluster tips and thus mechanical instabilities. Furthermore, this negative $\Delta V$ could promote fluid circulation, which may provide a positive feedback on the reaction, while at $1.1\,\mathrm{GPa}$ fluids mobility is smaller, which helps to keep antigorite clusters stable. Our observations thus suggest that a negative volume change is actually more favourable to trigger dynamic shear failure, implying that fluid overpressures only play a secondary role during embrittlement, as previously deduced from an experimental study on dehydrating serpentinized peridotite[13].

We propose an alternative model to dehydration-embrittlement, in which dehydration-driven stress transfer, rather than fluid overpressure, is the trigger of embrittlement. The conceptual model described below (Fig. 9) argues that dehydration-driven dynamic rupture in olivine-antigorite aggregates constitutes a 'percolation' problem.

In the model, dynamic shear failure occurs primarily in strong olivine sub-volumes surrounded by weak, dehydrating antigorite clusters. During dehydration, stress in antigorite crystals drops to zero, that is, they lose their load-bearing capacity. The resulting stress transfer to the surrounding olivine matrix induces a mechanical instability in the aggregates. The stress transfer is more critical when $\Delta V < 0$ because it promotes tensile stress concentrations at dehydrating antigorite cluster tips. The model also needs to take into account fracture energy and fracture nucleation length[35–37], along with grain size, olivine-antigorite fraction, antigorite-network connectivity and

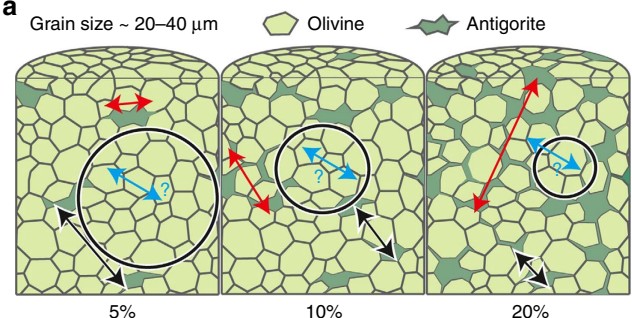

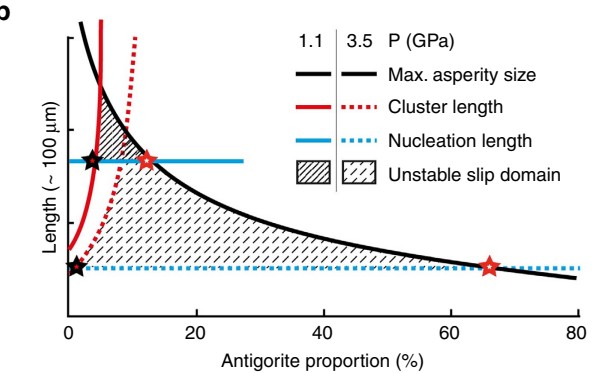

**Figure 9 | Conceptual model of dehydration-driven stress transfer faulting.** (**a**) Representative elementary volume showing the size of olivine sub-volumes as varying with antigorite content (black circles). All characteristic lengths are represented by arrows: average size of the olivine sub-volume ($l_{\mathrm{ol}}$, black), the critical fracture nucleation length ($l_{\mathrm{c}}$, blue) and the maximum antigorite cluster size ($\xi$, red); (**b**) dynamic shear failure is expected for critical antigorite fractions large enough to transfer a critical amount of stress (black star) and small enough for olivine asperities to be larger than $l_{\mathrm{c}}$ (red star), which is expected to decrease with increasing confining pressure.

pressure-temperature conditions. These parameters and their dimensions are detailed in the following paragraphs.

Olivine sub-volumes are bounded by antigorite grains. For a homogeneous distribution, the length of the sub-volumes is approximately equal to the mean spacing between antigorite grains, which depends both on the antigorite fraction ($\phi_{atg} = V_{atg}/V$, where $V_{atg}$ is the total antigorite volume and $V$ is the sample volume) and on the grain size. For spherical, isolated antigorite grains homogeneously distributed inside an olivine matrix, the average size of antigorite-free olivine sub-volumes is $l_{ol} = 2r[(1 - \phi_{atg})/\phi_{atg}]^{1/3}$, with the average antigorite-grain radius $r$ equal to $20\,\mu m$ in this study.

We define clusters of connected antigorite. Using a result from percolation theory[38], the length of antigorite clusters can be approximated by: $\xi \sim \beta |\phi_{atg} - \phi_{atg}^c|^\gamma$, where $\beta$ and $\gamma$ are geometrical parameters and $\phi_{atg}^c$ the critical antigorite fraction above which antigorite is connected at a certain scale. At this threshold, antigorite connectivity and olivine connectivity compete in reducing or increasing the probability for olivine sub-volumes to bear the stress or to yield. The parameter $\gamma$ is given by percolation theory to be 0.875 (ref. 38). For our modelling we assume $\beta = 0.118\ r$ (Fig. 9b). The parameter $\beta$ actually subsumes the role of grain size, homogeneity and geometry of the biphasic distribution. The calculation gives $\phi_{atg}^c \sim 6\,vol\%$ at 1.1 GPa and $\sim 18\,vol\%$ at 3.5 GPa (Fig. 9b).

For a homogeneous material, the critical nucleation length can be calculated from Griffith's rupture criterion[35,36] by: $l_c = \mu G/\Delta\sigma^2$, where $G$ is the Griffith's fracture energy ($\approx 1\,J\,m^{-2}$), and $\Delta\sigma$ is the stress drop ($< 100\,MPa$), which, at first order, can be considered proportional to the normal stress acting on the fault plane, and as such is also proportional to confining pressure. The nucleation length $l_c$ is expected to be at least $0.8\,\mu m$.

On the basis of rupture nucleation length and percolation theory[35–38] (Fig. 9), dynamic failure only occurs when: antigorite clusters are (1) small enough for the olivine sub-volumes to be larger than the critical nucleation length $l_c$ (minimum length for a fracture to start propagating dynamically) and (2) large enough to transfer a critical amount of load to the connected olivine skeleton. The elastic strain energy stored in the material depends on the relative sizes of weak antigorite clusters and load-bearing olivine sub-volumes. In other words, a critical antigorite content $\phi_{atg}^{c,max}$ exists (Fig. 9b), above which the characteristic length of olivine sub-volumes $l_{ol}$ becomes smaller than $l_c$. Moreover, the connectivity of antigorite clusters also affects the extent to which the olivine sub-volumes can store (and eventually release) significant elastic energy (that is, experimentally detectable with our AE setup). Below a second critical $\phi_{atg}^{c,min}$ value (Fig. 9b), antigorite grains are isolated and contribute too little to the overall load bearing for the stress-transfer mechanism to induce failure when dehydration commences.

Finally, $\phi_{atg}^{c,max}$ is expected to increase with confining pressure, $P_c$, at least at the laboratory scale (Fig. 9b), because $l_c$ inversely scales with stress drop. Thus the model predicts that, the higher the pressure, the smaller the size of potentially seismogenic sub-volumes. Smaller olivine sub-volumes would potentially become seismogenic at higher pressures, an observation supported by our experiments (Figs 2,3 and 5). In other words, at a given pressure, $\phi_{atg}^{c,min}$ is defined (black star, Fig. 9) above which the sub-volume is small enough for parts of the load-bearing network to break. Dehydrating antigorite clusters behave as fractures and induce high stress intensities in surrounding olivine sub-volumes. Above $\phi_{atg}^{c,max}$ (red star, Fig. 9), antigorite connectivity is too high for the olivine sub-volumes to store and release elastic energy, that is, the antigorite clusters are too large and numerous for the olivine load-bearing skeleton to be defined.

Our results reconcile previous experimental studies on dehydrating serpentinites, which reported conflicting 'seismic'[12–14] and 'aseismic' observations[14–19]. We found the occurrence of dynamic failure documented by acoustic emissions in experiments to depend on the degree of serpentinization (that is, volumetric ratio of olivine to serpentine) and pressure conditions[14]. Our observations also point towards a favourable role of negative volume changes for triggering dynamic failure on dehydration. This important observation, which was already reported by previous experimental studies[13,14], is not compatible with the typical dehydration-embrittlement model. We presented an alternative conceptual model, in which dehydration-driven stress transfer, rather than fluid overpressure, is the trigger of embrittlement. Our model incorporates antigorite distribution, volume ratio, grain size and fracture energy.

Our observation that 5 vol% dehydrating antigorite suffices, even at 3.5 GPa, to destabilize an antigorite-olivine aggregate also provides an explanation for intermediate-depth seismicity even for limited hydration of subducting lithospheric mantle[2,4,14]. In addition, our model provides a possible scaling of our laboratory observations to the field, since it relies on three up-scalable length scales. Fracture energy, and thus the nucleation length, scales with the size of seismic asperities[27]. In the laboratory, olivine- and antigorite-cluster-lengths scales depend on the grain size and the olivine-antigorite proportions. In nature, it is quite likely that deep mantle hydration occurs along pre-existing fault zones only[10,11]. Unaltered peridotite bodies, with volumes of the order of $1–100\,km^3$, bounded by discrete fault zones at various degrees of serpentinization[10,11] may constitute the equivalent of our olivine sub-volumes. The Lower Wadati–Benioff Plane may thus not solely correspond to a stability limit of antigorite[4,5], but also to a degree of hydration, that is, fault spacing and serpentinization extent.

Finally, our laboratory intermediate-depth earthquake analogues were observed at conditions where thermodynamic characteristics of the involved mineral reactions predicts that fluid overpressures are unlikely. Our observations of micro-pseudotachylytes produced under upper mantle conditions in millimetric samples suggests that dehydration-driven stress transfer, rather than fluid overpressure, leads to dynamic shear fracture nucleation, shear-induced melting, fault lubrication and complete, but transient, stress drops, for displacements of a few microns only. Our conclusions extend at much greater depth that of recent experimental studies[31–33] performed at low pressure, which highlighted the role played by flash melting in pure serpentinite.

## Methods

**Sample preparation.** Pure San Carlos olivine and Corsica antigorite were ground separately in an agate crusher. The powders were then sieved twice, respectively for the grain size not to exceed $40\,\mu m$ and for the mixture to be homogeneous. Grain size was checked using an optical microscope. Mixed powders were prepared with 5, 20 and 50 vol% antigorite fractions, during the second sieving.

The powders were encapsulated in gold jackets and hot-pressed in a piston-cylinder apparatus at 773 K over 1.5 GPa. Three samples per run were sintered. Samples display homogeneous distribution of antigorite, with final grain size ranging from 4 to $40\,\mu m$ (Fig. 1a).

Cylindrical samples of 3 mm length and 2.1 mm diameter were inserted into the D-DIA deformation assembly (Fig. 1b) at GSECARS (Advanced Photon Source, Chicago University). A hexagonal Boron Nitride (hBN) jacket prevents the sample from contacting the graphite furnace. X-ray diffraction patterns (51 keV), display intensities which are consistent with the target composition (Fig. 1d). The hBN jacket, the graphite furnace, and the Boron-epoxy cube are transparent to X-rays and give several peaks on the X-ray diffraction pattern (Fig. 1d).

**Sample deformation.** In this study, samples were deformed using a D-DIA apparatus equipped with an acoustic recording system[21]. The whole system is currently located in a synchrotron (APS, Chicago), where X-rays allow for *in situ* measurements of strain, stress and reaction progress. Two of the six anvils are

composed of sintered diamond, which are transparent to X-rays; the four other anvils are made of tungsten carbide (WC). Truncations of 6 mm were used. The samples were first compressed to 1.1 or 3.5 GPa, then heated to 773 K, and deformed with a strain rate of $5.0 \pm 0.1 \times 10^{-5}\,s^{-1}$. After 10% of strain, temperature was ramped at about $0.1\,K\,s^{-1}$. Table 1 summarizes the experimental conditions.

**X-ray data processing.** X-ray radiography allows monitoring the length of the sample, $l$, equal to the distance between the gold foils, which absorb X-rays. Axial strain is calculated from current length according to $\varepsilon = (l_0 - l)/l_0$, where $l_0$ is the initial length.

Details on X-ray processing can be found in previous publications[21,39]. Stresses are extracted from X-ray diffraction data analysis[40]. Two-dimensional powder X-ray diffraction patterns give access to the lattice strain experienced by crystallographic planes depending on their orientation relative to the maximum compressive axis. Stress in crystals is inferred from lattice strains[15,21,39,40], recovered from the X-ray diffraction (Fig. 1c,d) using the software package Multifit-Polydefix[41]. The diffracting plane azimuth $\phi$ is defined as the angle between the maximum compression axis and the diffracting plane normal, with $\phi = 0$ for planes normal to the maximum compression axis. The projection of $\phi$ on the detector plate is $\delta$, with $\cos\phi = \cos\theta \times \cos\delta$, where $\theta$ is the diffraction angle. Assuming absence of lattice preferred orientation, plastic relaxation of stress, and under an axial-symmetric stress field, elastic theory predicts the relation between measured d-spacing $d_{(hkl)}$ and lattice strain $Q_{hkl}$ to be[40,42]:

$$\frac{d_{(hkl)(\phi)} - d_{P(hkl)}}{d_{P(hkl)}} = Q_{hkl}\left(1 - 3\cos^2\phi\right) \qquad (6)$$

where $d_{P(hkl)}$ is the d-spacing corresponding to the mean stress part of the full stress tensor. The hydrostatic pressure is then calculated from these $d_{P(hkl)}$ using the equation of state of San Carlos olivine[43]. The stress is calculated as: $\tau_{hkl} = 6Q_{hkl} \times G_{hkl}$, where $G_{hkl}$ are effective moduli. The $G_{hkl}$ are calculated with elastic compliances $S_{ij}$ from inversion of the pressure and temperature dependent stiffness tensor $C_{ij}$ (refs 44–46). We used information from the 020, 021, 002, 130, 131 and 112 diffraction peaks of olivine, which were then employed to calculate average micro-stress values (Fig. 3). We assume that the average stress recorded by these planes corresponds to macroscopic stress.

**Recording and processing of acoustic emissions.** A piezoceramic transducer (Boston Piezo, PZT 5A, Tab coaxial, 10 mm diameter, 0.5 mm thick, 2 MHz center frequency) is glued behind each of the six anvils. Transducers have non-linear sensitivity but seem to work well within the 0.1–4 MHz frequency band. Raw acoustic data are amplified at 60 dB.

Acoustic emissions (AEs) are recorded using a trigger logic, such that 3–5 transducers need to reach a critical voltage (150 mV) within a time window of 50 ms to trigger recording. Sampling frequency of the digital oscilloscope is 50 MHz. The maximum recording rate, never reached during these experiments, is ~15–60 AEs events per second.

The setup is used to determine whether an acoustic emission originates within the sample volume or not. To do so, arrival times at facing anvils are compared to one another. If these are such that the difference, after accounting for the difference in slowness for tungsten carbide and sintered diamond, is $|\Delta t_{anvil}| < 0.25\,s$—a criterion which corresponds to an average P-wave velocity of $8\,km\,s^{-1}$ inside the sample—the corresponding AE is considered as coming from inside the sample. However, we are not interested in its precise location and lack the resolution to precisely associate microfaults with one event or another. More information about the location protocol can be found in the litterature[21].

The magnitude of an AE was calculated as: $M_{AE} = \log_{10}E$, with $E$ taken as the average energy ($V^2$) of the amplified acoustic signals. This quantity is relative, like the magnitude defined for earthquakes, and permits to compare one event with another. Magnitudes for 79 AEs were calculated. The magnitude distribution follows the Gutenberg-Richter power law with $\log_{10}N = 1.973 - 0.58\,M_{AE}$ (Fig. 4a). Here, $\log_{10}N_0 = 1.973$ or $N_0 = 94$, that is, our catalogue is almost complete and >80% of the dehydration-induced AEs generated during our experiments were recorded.

Figure 4b shows Fast Fourier transforms of waveforms for three AEs recorded behind a WC anvil representative of large, medium and small relative magnitude ($M_{AE} \approx 2.7, 1.1$ and $0.7$) during experiment 05–1.1 ($\phi_{atg} = 5$ vol%, $P_c \sim 1.1$ GPa). For each event, corner frequencies are found close to 2.5 MHz on average, irrespective of the event magnitude. It is possible that we are technically limited by our set-up in recording frequencies higher than this value (transducers, amplifiers, glue, as part of the whole receiver function) so that the corner frequency we compute constitutes a lower bound for real corner frequencies of these events. Considering a rupture velocity $v_r$ equal to the shear-wave velocity ($5\,km\,s^{-1}$), an upper bound to fault size results to $L \sim v_r/f_c = 2$ mm, matching sample dimensions.

**Data availability.** Raw data were generated at the Advanced Photon Source of the Argonne National Laboratory (Illinois). Derived data supporting the findings of this study are available from the corresponding author on request.

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

## Acknowledgements

We thank Dominique Badia-Fuchs, Christian Chopin, Jean-Noël Rouzaud, Yves Pinquier, Sebastien Merkel, Frank Bettenstedt and many others for their help and support. This research was funded by L'Agence Nationale de la Recherche (project 'DELF' ANR12-JS06-0003 to A.S.), the LABoratoire d'EXcellence MATISSE (UPMC—Sorbonne Universités). GeoSoilEnviroCARS is supported by the National Science Foundation–Earth Sciences (EAR-1128799) and U.S. Department of Energy–Geosciences (DE-FG02-94ER14466). Use of the Advanced Photon Source was supported by the U.S. Department of Energy, Office of Science, Office of Basic Energy Sciences, under contract DE-AC02-06CH11357. Y.W. acknowledges NSF support EAR-1361276. We thank ERC REALISM Grant # 681346.

## Author contributions

The project was developed by T.P.F., A.S., N.H., L.L. and H.W.G. II. T.F., S.I. and J.R. produced and provided sintered samples. T.F., S.I., N.H., A.S., Y.W., J.G. and L.L. carried out the deformation experiments. D.D. and T.F. provided SEM and TEM pictures. T.F. processed both acoustic and X-ray data and proposed interpretations, including the conceptual model.

## Additional information

**Competing interests:** The authors declare no competing financial interests.

