## [Peer Review File · Nature Communications]

Reviewers' Comments:

Reviewer #1 (Remarks to the Author)

The authors present experimental results on the role of dehydration on unstable fault growth in peridotite. These novel new experiments help to resolve apparently contradictory results on the role of dehydration reactions - and will be of interest to a wide range of scientists working on the dynamics of subduction zones and earthquakes processes. The data support previous work that indicates the unstable fault growth is not directly promoted by dehydration of antigorite. Overall I was impressed by the work presented in the paper, however, before the paper is published the authors need to explicitly address a few important points.

1. Most importantly, for experiments at 3.5 GPa, the authors observe essentially the same behavior with and without the presence of antigorite.

a. At 3.5 GPa, the onset of AE occurs at the same temperature for the 100% olivine sample and samples with antigorite (Figure 2).

b. At 1.1 GPa, it's intriguing that AE are observed primarily after total dehydration of the "soft" hydrous phases (i.e., antigorite and "talc-like") - thus in a rock with only olivine and pyroxene.

One issue that could be at play here is the "frictional stability" of the products versus reactants. At low pressure, the AEs are only prevalent in the 5% antigorite sample, but after the hydrous phases are dehydrated (i.e. when only frictionally unstable phases remain). In the higher pressure experiments, AE are detected at much lower temperature (with and without antigorite). Did the authors conduct an experiment to high strain in a sample within the antigorite stability field? (i.e. at a T below the "talc-like" phase is seen)? Would AE be observed? In this context, the observation that antigorite becomes more "brittle" (showing more localized brittle behavior and less velocity strengthening frictional behavior) at high T and P may be relevant (c.f. Proctor and Hirth, JGR, 2016).

2. The authors' comments on the temperatures where the dehydration reactions occur is confusing. The authors note that antigorite dehydrates at somewhat higher temperature than predicted by the phase diagrams. Could this simply relate to the difference in the amount needed for detection in XRD relative to that needed to observed reaction products in retrieved samples in the kinetic experiments? At same time, Figure 1 shows that they observe the formation of the "talc-like" phase (involving dehydration of antigorite) at lower T than the kinetic studies. The latter observations seems inconsistent with the temperature they claim they see evidence for antigorite dehydration?

3. It would be helpful for the authors to compare their microstructural results to the recent work of Brantut et al (Geology 2016) on dynamic versus static stress drops during dynamic slip in antigorite.

Reviewer #2 (Remarks to the Author)

Review to the manuscript: "Dehydration-driven stress transfer triggers intermediate-depth earthquakes"

This is an interesting paper conducting a series of acoustic emission measurements on olivine/antigorite aggregate during syndeformational dehydration conditions of antigorite at high pressure. Lower Wadati-Benioff seismic plane within the subducting mantle is formed at 550-800°C, conditions where the dehydration reaction of antigorite serpentine occurs, thus dehydration reactions involving antigorite serpentinite breakdown have been suggested to promote

seismicity on the lower Wadati-Benioff plane. Although the phenomenon of dehydration embrittlement of serpentine was discovered almost 50 years ago (Raleigh and Paterson, 1965), it had still been controversial whether the dehydration reaction of serpentine really triggers earthquakes or not. Some recent experimental works showed the lack of instability during the dehydration of "pure serpentinite" (e.g., Gasc et al., 2011, Chernak and Hirth, 2011, Okazaki and Hirth, 2016). This work demonstrates that dehydration reaction of "partially serpentinitized peridotite" can be associated with seismicity but not pure serpentinite. They propose a conceptual model triggering the unstable slip by a dehydration reaction with negative volume change; the dehydration reaction of antigorite induces a stress concentration into strong (i.e., seismic) olivine clusters. The result is exciting, of wide interest not only for rock physicists but also seismologists. Within the citation limit this paper seems to have the major relevant literatures.

Overall the paper is well put together and convincing, and certainly worth publishing in Nature Communications. I have only several minor comments and questions. Hope my comments help to strengthen the manuscript.

1. One of my questions is why any AEs were not observed during the breakdown of antigorite at the lower pressure condition. Actually, AEs were observed during 'talc-like' phase breakdown at the lower pressure condition. Antigorite breakdown itself doesn't seem to radiate AE at low pressure. If the authors' model is correct, unstable slip should be triggered by breakdown of both antigorite and talc-like phase, so AE activity should be widely observed above the temperature of 823K. Is this correct? On the other hand, the purpose of this research is to understand the mechanism triggering intermediate-depth earthquakes. In terms of that, which reaction induces the seismicity is not very important because both antigorite and talc dehydrations are in the same series of reactions of hydrated peridotite within the limited temperature range (550-800°C) corresponding to temperatures on the lower Wadati-Benioff zone in the subducting mantle. It would be nice if authors mention those in the manuscript.

2. Is the dehydration reaction of antigorite really necessary to induce AEs? It is a function of the antigorite proportion in the manuscript and a stress concentration in olivine clusters may occur without dehydration reaction because of a huge strength difference between olivine and antigorite (e.g., Ref. 24 in the manuscript). Experiments shown in Fig. 2 observe no AE before the temperature ramping, but the amount of total strain (~10%) may not be enough to reach a steady-state flow based on the result by Chernak and Hirth (2011). It could be nice if the authors conduct one or two extra experiments using same starting sample at same pressure condition but without temperature ramping.

3. The 100% olivine experiment at the higher pressure condition also observed AEs during the temperature ramping. This might suggest that a weakening of strong (brittle) material is the key to induce seismicity by whatever mechanisms, e.g., dehydration weakening by releasing the pore pressure and/or thermal weakening of the flow strength (i.e., dislocation/diffusion creep strength of olivine).

4. Line 3 on the caption of Fig. 1: geotherms of where? Slab surface, Moho or the lower Wadati-Benioff plane?

5. Line 4 in Chapter 6 in Supplementary Information: Replace the chemical formula for forsterite from $Mg_2Si_4O_8$ to Mg_2SiO_4 .

Reviewer #3 (Remarks to the Author)

Summary of the key results

The manuscript reports new state-of-the-art results deploying a high-pressure multi-anvil press experiment at a synchrotron facility. On top of it, the manuscript presents a detailed description of

experimental products using textual analysis with TEM and SEM, and seismic energy analysis from the observed waveforms. I think the manuscript presents an unparalleled analysis in experimental samples for acoustic emission.

The manuscript presents a new hypothesis that can reconcile contradictory experimental results as well as a cause of natural seismic phenomena. It focuses on identifying the mechanism that causes seismic events inside the descending subduction plates. These earthquakes have been known for probably nearly 100 years, but their cause is always debated.

The manuscript's claim is that the right amount of serpentine within an olivine aggregate can create a stress-loading on a rigid olivine grains. Upon the serpentine dehydration, the released fluid is sufficient to perturb the stress-supporting network, and causes fractures (i.e. experiment-scale earthquakes). It is a new hypothesis and certainly an attractive solution to a well-debated issue.

Originality and interest: if not novel, please give references

As I have stated, the problem itself, the cause of "intermediate-depth, within-the-plate, seismic events", has been known for many many years. Also, there have been number of high pressure deformation experiments with specific focus on the acoustic emissions (e.g. Dobson et al. 2002 Science). Neither the problem nor the experiment is original. Rather, the original aspect of the manuscript is the new model that can explain the cause of the seismic events.

Data & methodology: validity of approach, quality of data, quality of presentation

It appears to me, the experimental method, and acquired data are all appropriate. However, given the nature of data acquired from diverse methods, and complexity of related discussions, I must request authors to provide improved data presentations than what is given here. It was simply impossible to evaluate the arguments presented in the main text without consulting the supplementary document. I wonder if a short manuscript format would provide a proper platform present such complex discussions to the readers. As minimum, I just have to say that it was not a smooth reading to understand the construction of the logic between experimental results and conclusions.

Appropriate use of statistics and treatment of uncertainties

A total of eight experiments were conducted and 72 acoustic signals are capture. I doubt these are sufficient number of data to provide statistical test.

Conclusions: robustness, validity, reliability

There are two main conclusions in this manuscript. I would comment independently. Would the model proposed by the authors explain the cause of natural phenomena? I have to say "maybe", the manuscript presents a potential model, certainly very attractive one, but it is not the only mechanism proposed for the cause of such earthquakes. Furthermore, I am not fully convinced that this study has explored all parameters. For example, the strain rate of the deformation is kept constant. While the manuscript provides compelling scaling argument for the choice of the strain rate, I think it is unlikely that nature is represented by a single strain rate. The model of critical serpentine fraction to cause a acoustic emission. This is conclusion is very well demonstrated with multitude of observation presented here. It is certainly intriguing to see the authors' effort to apply quantitative arguments to various observations. This part of the paper is certainly new, and shows an original model well-supported by observations.

Suggested improvements: experiments, data for possible revision

As I have complained, this is a difficult paper to understand the construction of the discussions. Certainly, the manuscript's main text reads rather like a series of brief summaries, and it is impossible to evaluate scientific soundness and validity of arguments just from the main text. To appreciate the extent of the work that went behind, the reader is required to read a 25-page electronic supplementary document, which is also difficult to read. I must say that the manuscript attempts to pack so many arguments based on a wide range of data, it simply is an extended summary at the current format. I would just like to ask authors if a short format paper is the best way to present a high quality work presented here.

If authors insist on the short format paper, I would suggest to reorganise the paper to construct the main discussion using one or two key observations, rather than flooding the main text with many dispersed arguments. This way, there should be enough room to develop complete discussions.

Dehydration-driven stress transfer triggers intermediate-depth earthquakes – point by point response to reviewers

→ Answers are in bold type.

Reviewer #1 (Remarks to the Author):

The authors present experimental results on the role of dehydration on unstable fault growth in peridotite. These novel new experiments help to resolve apparently contradictory results on the role of dehydration reactions - and will be of interest to a wide range of scientists working on the dynamics of subduction zones and earthquakes processes. The data support previous work that indicates the unstable fault growth is not directly promoted by dehydration of antigorite. Overall I was impressed by the work presented in the paper, however, before the paper is published the authors need to explicitly address a few important points.

1. Most importantly, for experiments at 3.5 GPa, the authors observe essentially the same behavior with and without the presence of antigorite.

a. At 3.5 GPa, the onset of AE occurs at the same temperature for the 100% olivine sample and samples with antigorite (Figure 2).

This is an excellent point. Unfortunately, in the former version formatted as a short letter, space limitations did not allow us to discuss it in details. In this new version, complementary pictures (Fig.5d) are provided and highlight that the brittle mechanism occurring in pure olivine samples is not the same as the one occurring in antigorite-bearing samples. This is now extensively discussed in the core of the manuscript:

“The pure-olivine sample deformed at 3.5 GPa (Fig. 5d) exhibits an intriguing microstructure, with shear bands, about 50-100 μm thick, crosscutting the sample. These shear bands are filled with angular micro to sub-micrometric olivine grains. However, note that in this antigorite-free sample, AEs were less numerous, detected at the highest macroscopic deviatoric stress only, which also corresponded to the beginning of temperature ramping (Fig. 3). This might point towards a possible role played by thermal stress relaxation in olivine crystals (or the assembly). Finally, large cataclastic shear bands are rarely associated with a single dynamic (rapid) shear failure, because the elastic energy is converted into fracture surface energy dissipated in the shear zone rather than radiated at high speed. It is also possible that further heating and deformation overprinted the actual microstructure upon failure. No further investigations of the shear bands observed in this single pure olivine sample have been performed” (lines 157-168).

b. At 1.1 GPa, it's intriguing that AE are observed primarily after total dehydration of the "soft" hydrous phases (i.e., antigorite and "talc-like") - thus in a rock with only olivine and pyroxene.

We are also intrigued by this experimental result and have two possible explanations to rationalize this observation:

I. the X-ray beam only illuminates a portion of the sample (200x200 microns typically) while the sample is mm sized. The “out” boxes (Fig. 2, previous Fig. 1) indicate the limits above which we do not detect phases any more. There may be some atg remaining in other portions of the sample. Note this limit is higher in samples with higher atg content, as the probability to find it in the beam path is higher.

II. we now comment the fact that at 1.1 GPa *“AEs were observed [...] only with low antigorite fraction (5 vol.%), and only after the breakdown of talc, i.e. a reaction associated with a negative volume change (Fig. 2), [...] suggesting that a negative volume change is actually more favorable to trigger dynamic shear failure. This has an important consequence, because it implies that fluid overpressures only play a secondary role during embrittlement, as was previously suggested by experimental study on serpentinized peridotite”* (lines 213-223).

One issue that could be at play here is the "frictional stability" of the products versus reactants. At low pressure, the AEs are only prevalent in the 5% antigorite sample, but after the hydrous phases are dehydrated (i.e. when only frictionally unstable phases remain). In the higher pressure experiments, AE are detected at much lower temperature (with and without antigorite). Did the authors conduct an experiment to high strain in a sample within the antigorite stability field? (i.e. at a T below the "talc-like" phase is seen)? Would AE be observed? In this context, the observation that antigorite becomes more "brittle" (showing more localized brittle behavior and less velocity strengthening frictional behavior) at high T and P may be relevant (c.f. Proctor and Hirth, JGR, 2016).

We understand the point raised by the reviewer. Experiments realized by Gasc, Hilairat, Wang, Yu, Ferrand and Schubnel (unpublished results, publication in preparation), at 3.5 GPa, with the same experimental setup and protocol (thus moderate strains) on the same antigorite indeed seem to show moderate AE activity in the antigorite stability field ~ 400°C. However, we believe this is beyond the scope of the present study.

2. The authors' comments on the temperatures where the dehydration reactions occur is confusing. The authors note that antigorite dehydrates at somewhat higher temperature than predicted by the phase diagrams. Could this simply relate to the difference in the amount needed for detection in XRD relative to that needed to observed reaction products in retrieved samples in the kinetic experiments? At same time, Figure 1 shows that they observe the formation of the "talc-like" phase (involving dehydration of antigorite) at lower T than the kinetic studies. The latter observations seems inconsistent with the temperature they claim they see evidence for antigorite dehydration?

We apologize for this confusion and clarified the text:

“In our experiments, antigorite dehydration was observed between 873 and 973 K at 1.1 GPa and between 873 and 923 K at 3.5 GPa (Fig. 2). Onsets of reactions 1 (“tl-in”) and 2 (“Atg –”), were observed at a slightly lower temperature than in kinetics studies (Perrillat et al., 2005; Chollet et al., 2011). However, due to our temperature calibration method on undeformed cells assemblies, temperature might be underestimated during deformation. Indeed, sample shortening along the compression axis reduces the distance between the top and bottom WC anvils, which are efficient heat sinks (Raterron et al. 2013).

On the contrary, reaction products were detected by XRD at temperatures higher than those determined by kinetics studies (Perrillat et al., 2005; Chollet et al., 2011): a talc-like phase appeared between 723 and 923 K irrespective of pressure and enstatite was only detected at temperatures higher than 953 K at 1.1 GPa (Fig. 2). These oversteps in temperature, which can be attributed to the fast heating rate, might be caused by several factors, including reaction kinetics, in particular that of grain nucleation/growth. Finally, a faster decrease of the antigorite diffraction peaks (Fig. 2, "Atg -") indicates an acceleration of antigorite dehydration, thus promoting the 1st reaction completion and the production of the "talc-like" phase." (lines 120-133).

Let us also note that dehydration of a partially connected network of hydrous phase depends on many parameters, especially the possibility for water to escape, e.g. grain size, radius of connections, pressure. The "tl-in" box in Fig. 2 (initially Fig. 1) highlights the onset of reaction 1 where atg reacts only partially according to diffraction patterns. It is actually pretty consistent with previous studies (eg. Perrillat et al, 2005). The "Atg - " box corresponds to the time at which we see a faster decrease of the antigorite diffraction peaks and is the onset of reaction 2. Actually, reaction 2 consumes the "talc-like" phase and thus promotes the 1st reaction completion. Reactions 1 (at 3.5 GPa) and 2 (both pressure sets) are indeed longer to occur here, compared to previous studies. This might be due to several factors including kinetics of reaction (the heating rate and amount are moderate).

Besides, for the "tl-in" and "Atg-" boxes, the T gradient also increases in the cell due to shortening of the sample, cell material along the compression axis and therefore of the distance between top and bottom WC anvils, which are actually a heat sink (e.g. Raterron et al 2013, on a smaller cell design). Therefore the calculated T may be higher than the actual T in the cell, and here we rely on the diffraction observations, which is unambiguous information.

3. It would be helpful for the authors to compare their microstructural results to the recent work of Brantut et al (Geology 2016) on dynamic versus static stress drops during dynamic slip in antigorite.

This is an excellent point and this comparison is now in (new) section "Microstructural observations" (lines 190-196, ref. 31).

Reviewer #2 (Remarks to the Author):

Review to the manuscript: "Dehydration-driven stress transfer triggers intermediate-depth earthquakes"

This is an interesting paper conducting a series of acoustic emission measurements on olivine/antigorite aggregate during syndeformational dehydration conditions of antigorite at high pressure. Lower Wadati-Benioff seismic plane within the subducting mantle is formed at 550-800°C, conditions where the dehydration reaction of antigorite serpentine occurs, thus dehydration reactions involving antigorite serpentine breakdown have been suggested to promote seismicity on the lower Wadati-Benioff plane. Although the phenomenon of dehydration embrittlement of serpentine was discovered almost 50 years

ago (Raleigh and Paterson, 1965), it had still been controversial whether the dehydration reaction of serpentine really triggers earthquakes or not. Some recent experimental works showed the lack of instability during the dehydration of "pure serpentinite" (e.g., Gasc et al., 2011, Chernak and Hirth, 2011, Okazaki and Hirth, 2016). This work demonstrates that dehydration reaction of "partially serpentinitized peridotite" can be associated with seismicity but not pure serpentinite. They propose a conceptual model triggering the unstable slip by a dehydration reaction with negative volume change; the dehydration reaction of antigorite induces a stress concentration into strong (i.e., seismic) olivine clusters. The result is exciting, of wide interest not only for rock physicists but also seismologists. Within the citation limit this paper seems to have the major relevant literatures.

We thank the reviewer for reminding us of the reference by Raleigh and Paterson, which should indeed be cited and was added to the revised manuscript (line 27, ref. 5).

Overall the paper is well put together and convincing, and certainly worth publishing in Nature Communications. I have only several minor comments and questions. Hope my comments help to strengthen the manuscript.

1. One of my questions is why any AEs were not observed during the breakdown of antigorite at the lower pressure condition. Actually, AEs were observed during 'talc-like' phase breakdown at the lower pressure condition. Antigorite breakdown itself doesn't seem to radiate AE at low pressure. If the authors' model is correct, unstable slip should be triggered by breakdown of both antigorite and talc-like phase, so AE activity should be widely observed above the temperature of 823K. Is this correct? On the other hand, the purpose of this research is to understand the mechanism triggering intermediate-depth earthquakes. In terms of that, which reaction induces the seismicity is not very important because both antigorite and talc dehydrations are in the same series of reactions of hydrated peridotite within the limited temperature range (550-800°C) corresponding to temperatures on the lower Wadati-Benioff zone in the subducting mantle. It would be nice if authors mention those in the manuscript.

We are also intrigued by this experimental result. However, we aim here to show that antigorite (or talc-like) dehydration itself cannot trigger dynamic shear failure, but that local destabilization of any hydrous phase around 600-800°C may lead to a stress transfer triggering mechanical instabilities in a stronger material. As a consequence, the reason why antigorite is so important is the correlation of its thermal breakdown with a sufficient stress level in nature. We also consider that the 'talc-like' phase as a transient, very likely metastable product during antigorite dehydration, which ultimately produces enstatite, olivine and water (also see Perrillat et al, 2005). Note that this 'talc-like' phase has never been observed in nature. We clarified this in the manuscript. (lines 120-133):

"In our experiments, antigorite dehydration was observed between 873 and 973 K at 1.1 GPa and between 873 and 923 K at 3.5 GPa (Fig. 2). Onsets of reactions 1 ('tl-in') and 2 ('Atg -'), were observed at a slightly lower temperature than in kinetics studies (Perrillat et al., 2005; Chollet et al., 2011). However, due to our temperature calibration method on undeformed cells assemblies, temperature might be underestimated during deformation. Indeed, sample shortening along the compression axis reduces the distance between the top and bottom WC anvils, which are efficient heat sinks (Raterron et al. 2013).

On the contrary, reaction products were detected by XRD at temperatures higher than those determined by kinetics studies (Perrillat et al., 2005; Chollet et al., 2011): a talc-like phase appeared between 723 and 923 K irrespective of pressure and enstatite was only detected at temperatures higher than 953 K at 1.1 GPa (Fig. 2). These oversteps in temperature, which can be attributed to the fast heating rate, might be caused by several factors, including reaction kinetics, in particular that of grain nucleation/growth. Finally, a faster decrease of the antigorite diffraction peaks (Fig. 2, 'Atg -') indicates an acceleration of antigorite dehydration, thus promoting the 1st reaction completion and the production of the 'talc-like' phase" (lines 120-133).

In addition, we discussed the effect of negative vs positive Clapeyron slope in the "Antigorite stability and thermal breakdown" section. Surprisingly, our data seems to show "a negative volume change is actually more favorable to trigger dynamic shear failure. This has an important consequence, because it implies that fluid overpressures only play a secondary role during embrittlement, as was previously suggested by experimental study on serpentinized peridotite" (lines 220-223).

2. Is the dehydration reaction of antigorite really necessary to induce AEs? It is a function of the antigorite proportion in the manuscript and a stress concentration in olivine clusters may occur without dehydration reaction because of a huge strength difference between olivine and antigorite (e.g., Ref. 24 in the manuscript). Experiments shown in Fig. 2 observe no AE before the temperature ramping, but the amount of total strain (~10%) may not be enough to reach a steady-state flow based on the result by Chernak and Hirth (2011). It could be nice if the authors conduct one or two extra experiments using same starting sample at same pressure condition but without temperature ramping.

Here we argue that the stress transfer, which is necessary to generate a mechanical instability, requires a drastic but local change in rheology. In other terms, when an antigorite cluster (with its own length, connectivity and geometry) dehydrates, the rheology contrast between the two phases shoots up.

Experiments have been performed on olivine aggregates for decades, investigating a large spectrum of temperature, pressure and strain rate, including by two of the authors of this ms. No brittle behavior within pure polycrystalline olivine was reported/observed in the conditions we used. Hence, in Fig. 3 (prev. Fig. 2), we consider the comparison between the experiments with 0 and 5 vol.% antigorite at 1.1 GPa as an evidence of antigorite being required for brittle behavior.

3. The 100% olivine experiment at the higher pressure condition also observed AEs during the temperature ramping. This might suggest that a weakening of strong (brittle) material is the key to induce seismicity by whatever mechanisms, e.g., dehydration weakening by releasing the pore pressure and/or thermal weakening of the flow strength (i.e., dislocation/diffusion creep strength of olivine).

We thank the reviewer for this insightful comment. This point is partially answered above (reviewer 1, point 1). Complementary pictures of the pure olivine sample (Fig.5d) are now provided in the revised manuscript. These highlight that the brittle mechanism occurring in pure olivine samples is different than the one occurring in antigorite-bearing samples. Thermal weakening may be at play here, in the highly stressed pure olivine sample and may be the

cause of the specific microstructure. These microstructures and stress level are consistent and allow us to affirm that, in any case, the mechanism triggering dynamic shear failure in antigorite-bearing and antigorite-free samples is different. This is now extensively discussed in the core of the manuscript:

“The pure-olivine sample deformed at 3.5 GPa (Fig. 5d) exhibits an intriguing microstructure, with shear bands, about 50-100 μm thick, crosscutting the sample. These shear bands are filled with angular micro to sub-micrometric olivine grains. However, note that in this antigorite-free sample, AEs were less numerous, detected at the highest macroscopic deviatoric stress only, which also corresponded to the beginning of temperature ramping (Fig. 3). This might point towards a possible role played by thermal stress relaxation in olivine crystals (or the assembly). Finally, large cataclastic shear bands are rarely associated with a single dynamic (rapid) shear failure, because the elastic energy is converted into fracture surface energy dissipated in the shear zone rather than radiated at high speed. It is also possible that further heating and deformation overprinted the actual microstructure upon failure. No further investigations of the shear bands observed in this single pure olivine sample has been performed.” (lines 157-168).

Second, here we show that pore pressure is not the major factor controlling peridotite embrittlement during antigorite dehydration. Dehydration embrittlement, i.e. involving pore overpressure, is not at play in the DDST (Dehydration-Driven Stress Transfer) model.

4. Line 3 on the caption of Fig. 1: geotherms of where? Slab surface, Moho or the lower Wadati-Benioff plane?

We thank the reviewer and apologize for being imprecise. This is now corrected in Fig. 2 caption (prev. Fig. 1) to : “CHZ/CSZ: geotherms on top of the slab in Cold/Hot Subduction Zone”.

5. Line 4 in Chapter 6 in Supplementary Information: Replace the chemical formula for forsterite from Mg₂Si₄O₈ to Mg₂SiO₄.

We thank the reviewer and apologize for this typo.

Reviewer #3 (Remarks to the Author):

Summary of the key results

The manuscript reports new state-of-the-art results deploying a high-pressure multi-anvil press experiment at a synchrotron facility. On top of it, the manuscript presents a detailed description of experimental products using textual analysis with TEM and SEM, and seismic energy analysis from the observed waveforms. I think the manuscript presents an unparalleled analysis in experimental samples for acoustic emission.

The manuscript presents a new hypothesis that can reconcile contradictory experimental results as well

as a cause of natural seismic phenomena. It focuses on identifying the mechanism that causes seismic events inside the descending subduction plates. These earthquakes have been known for probably nearly 100 years, but their cause is always debated.

The manuscript's claim is that the right amount of serpentine within an olivine aggregate can create a stress-loading on a rigid olivine grains. Upon the serpentine dehydration, the released fluid is sufficient to perturb the stress-supporting network, and causes fractures (i.e. experiment-scale earthquakes). It is a new hypothesis and certainly an attractive solution to a well-debated issue.

Originality and interest: if not novel, please give references

As I have stated, the problem itself, the cause of "intermediate-depth, within-the-plate, seismic events", has been known for many many years. Also, there have been number of high pressure deformation experiments with specific focus on the acoustic emissions (e.g. Dobson et al. 2002 Science). Neither the problem nor the experiment is original. Rather, the original aspect of the manuscript is the new model that can explain the cause of the seismic events.

Data & methodology: validity of approach, quality of data, quality of presentation

It appears to me, the experimental method, and acquired data are all appropriate. However, given the nature of data acquired from diverse methods, and complexity of related discussions, I must request authors to provide improved data presentations than what is given here. It was simply impossible to evaluate the arguments presented in the main text without consulting the supplementary document. I wonder if a short manuscript format would provide a proper platform present such complex discussions to the readers. As minimum, I just have to say that it was not a smooth reading to understand the construction of the logic between experimental results and conclusions.

We understand the remark and agree with the reviewer. The earlier version of this manuscript was probably confusing. We present here a revised manuscript that we hope is much easier to follow. All points are now in the main text, there is no more supplementary information.

Appropriate use of statistics and treatment of uncertainties

A total of eight experiments were conducted and 72 acoustic signals are capture. I doubt these are sufficient number of data to provide statistical test.

We reproduced twice one of the crucial experiments here (exp. 1619 and D1777, table 1) to make sure of the absence of AE during the dehydration on this composition, which is already an unusual care for synchrotron-based experiments, due to the limited time on these instruments. For sure more experiments could be performed. However due to the limited time, heaviness of preparing, carrying out and analyzing synchrotron experiments, we believe a true statistical study could be done rather using off-line high-pressure presses and analyzing post-mortem microstructures. Especially, a larger amount of experiments is likely to give access to

somewhat predictable fault length depending on the relative size of antigorite-free and antigorite-bearing bodies, first at laboratory-scale, then allowing numerical modeling and upscaling to natural features. But this is clearly beyond the scope of the present study.

Conclusions: robustness, validity, reliability

There are two main conclusions in this manuscript. I would comment independently.

Would the model proposed by the authors explain the cause of natural phenomena? I have to say "maybe", the manuscript presents a potential model, certainly very attractive one, but it is not the only mechanism proposed for the cause of such earthquakes.

We agree with the reviewer and did not intend to state this mechanism is the only one at play. In fact this is now clearly stated in our Introduction. Hopefully, future modeling will manage to combine our DDST with shear heating (Kelemen & Hirth, 2007) and self-localizing thermal runaway (John et al., 2009). We nevertheless argue that DDST is an essential ingredient in triggering mechanical instabilities at intermediate depth.

Furthermore, I am not fully convinced that this study has explored all parameters. For example, the strain rate of the deformation is kept constant.

We completely agree with the reviewer, but we deliberately chose the conditions we thought to be relevant according to the literature. Again, the experiments here are already a large dataset (more than 12 days of beamtime) given the limited time available on the instruments. This did not allow us to investigate more parameters, but obviously this could be the scope of future studies, potentially within a Griggs apparatus currently under installation at our home institute.

While the manuscript provides compelling scaling argument for the choice of the strain rate, I think it is unlikely that nature is represented by a single strain rate.

Again, this is an insightful comment. However, as stated in text, the ratio between the stress rate and the temperature increase is in the range of values calculated for real subducting slabs (Chernak & Hirth, 2011) and limited by experimental time, beam time allocation, etc., so that in the end we aimed at the best possible compromise. Finally, let us note that even if the sample-scale strain rate is constant in our developed experiment, the strain rate surely increases locally in the vicinity of dehydrating antigorite clusters.

The model of critical serpentine fraction to cause a acoustic emission. This is conclusion is very well demonstrated with multitude of observation presented here. It is certainly intriguing to see the authors' effort to apply quantitative arguments to various observations. This part of the paper is certainly new, and shows an original model well-supported by observations.

Suggested improvements: experiments, data for possible revision

As I have complained, this is a difficult paper to understand the construction of the discussions. Certainly,

the manuscript's main text reads rather like a series of brief summaries, and it is impossible evaluate scientific soundness and validity of arguments just from the main text. To appreciate the extent of the work that were behind, the reader is required to read a 25-pages electronic supplementary document, which is also difficult to read. I must say that the manuscript attempts to pack so many arguments based on a wide range of data, it simply is an extended summary at the current format. I would just like to ask authors if a short format paper is the best way to present a high quality work presented here.

If authors insist on the short format paper, I would suggest to reorganize the paper to construct the main discussion using on one or two key observations, rather than flooding the main text with many dispersed arguments. This way, there should be enough room to develop complete discussions.

We understood this point and propose here a new version without any supplementary information, everything being now in the main text. The paper is now cut in sections and hopefully the reviewer will find it easier to follow.

Reviewers' Comments:

Reviewer #1 (Remarks to the Author)

I am happy with the changes the authors made to address my (and others) comments. I recommend that the paper be published

Reviewer #2 (Remarks to the Author)

I read through the revised paper and the comments from the authors. The revised manuscript looks better than the original one, while I have a few comments relating to my first review.

1. Still the authors haven't answered my first question. The authors commented "local destabilization of ANY HYDROUS PHASE around 600-800°C may lead to a stress transfer triggering mechanical instabilities in a stronger material.", but obviously the destabilization of antigorite at 1.1 GPa doesn't induce any AE activity; only the destabilization of Tlc-like phase does in this condition. The authors' idea, DDST should be applicable for both reactions if it is true. The critical characteristic length either for l_c or ξ might be different in each reaction, but the authors didn't mention it. While temperature measurement has a huge uncertainty, in-situ XRD measurements strongly suggest this issue.

2. The author replied to my second comment in my first review:

"Experiments have been performed on olivine aggregates for decades, investigating a large spectrum of temperature, pressure and strain rate, including by two of the authors of this ms. No brittle behavior within pure polycrystalline olivine was reported/observed in the conditions we used. Hence, in Fig. 3 (prev. Fig. 2), we consider the comparison between the experiments with 0 and 5 vol.% antigorite at 1.1 GPa as an evidence of antigorite being required for brittle behavior."

but actually this manuscript reported the brittle behavior of the olivine aggregates at 3.5 GPa, it is very strange. Cumulative No. of AE of 100% olivine at 3.5 GPa is 30 to 50% less than those of antigorite/olivine mixtures (just 10 counts difference). Is this ok to say that it is the significant difference (line 165)?

Minor comment:

I should have said this in the first review, but in figure 2, it is better for this manuscript to plot the PT path of the Moho or both the Moho and the slab surface rather than the PT of the slab surface only. The PT path of the Moho seems to be more important for this research.

Reviewer #3 (Remarks to the Author)

Comments on the revised manuscript, "Dehydration-driven stress transfer triggers intermediate-depth earthquakes" submitted to Nature communications.

This is the second review of the manuscript after a revision responding to the comments made by three reviewers. As I have expressed in my first review, I was impressed with the data presented in the paper including variety of analytical method applied to experimental samples synthesized under high-pressure and high-temperature conditions. With such effort, the manuscript presents a new model explaining the cause of the intermediate-depth, lower Wadati-Benioff plane (LBP) earthquakes. I certainly appreciate the discussion and analytical effort invested in the paper. However, I was less impressed with the presentation of the subject. For the comments on the second review, I primarily focus my critiques on the presentation of the paper, as well as comments on the reply.

First of all, I had pleasant experience reading the second manuscript. I acknowledge the writing effort and that the increased number of SEM-TEM images (Figure 5,6,7,and) helped me follow the argument made by the authors. For example, I appreciate the new Figure 6 which displays two identical SEM images with and without a structural interpretation. Overall, the manuscript increased its length from eight to twenty pages main text. I believe the revised, longer, version is more accessible to a wider audience. I have not much comment to add with respect to scientific aspects of the paper.

I have made some comments on which mainly related to structure of the presentation and clarity/precision.

From line 204 to 234, the author deviate from result presentation under the heading of "microstructural observations," and start discussions related to the significance of microstructure observations. Perhaps, these paragraph should be moved into the discussion section.

I disagree with how some references are quoted, and the passage should be revised. line 34 - 38, here the reference 10 and 11 (Ranero et al. 2003 and Shillington et al. 2015) are cited to say, "documenting that the lithospheric mantle was partially hydrated tens of kilometers below the Moho". Ranger et al. 2003 shows the presence of faults with the seismic images covering approximately to 8 km below the MOHO. Then they argue for a possibility of extending such fault to 35 km depth based on an estimate of the brittle-ductile transition of the lithosphere.

According to Shillington et al. 2016, their seismic reflection data shows extensive hydration in the uppermost 3-4km of the mantle, not "tens" of kilometer. In addition to such observation, they argue that the correlation between the occurrence of LBP earthquakes and nature of brittle deformation at outer rise demonstrates the presence of hydrated (antigorite) layer, assuming that the intermediate earthquake is caused by antigorite.

For these reasons above, the citation of these papers as the demonstration of the presence of antigorite at the depth only provides an incomplete and circular argument, and furthermore misleading. As far as I am aware, there are no direct observation documenting the presence of serpentine at the 30 km depth below the MOHO. While I would not argue that there are no serpentine at the depth, but I would request the authors to present a leveled overview of the issue.

line 97-98, "can be stoichiometrically written as" The exact stoichiometry of antigorite is not what is written here. It is instructive to have an approximate chemical formula but please state that it is an approximate reaction stoichiometry. Antigorite stoichiometry is $M_{(3m-3)}T_{(2m)}O_{(5m)}(OH)_{(4m-6)}$ where $m = 14$ to 23 .

Line 193 : The issue raised by Reviewer 2, which is regarding a lack of an exact coincidence of the experimental result and the previous kinetic studies, may not significantly challenge the validity of the result presented here. It should be noted that the in situ kinetic experiments of Perrillat et al., 2005 and Chollet et al., 2011 have an issue of the temperature **accuracy** because their T calibration relies on the equations of state of internal markers. The offset pointed out by Reviewer 2 is well within the expected uncertainty of accuracy. Furthermore, I certainly agree with the authors that such offset can also be explained by the kinetic overstepping caused by heating strategy.

Line 153-155 : "Nevertheless, the three orders of magnitude covered by source sizes (Fig. 4a) suggest that our observations could be upscaled". It is certainly a great achievement to show that the same physical phenomenon can be scaled up to 3 orders of magnitude. However, that alone would not be a sufficient reason to suggest that the same is true for larger scale seismic event in

nature. Here, I would like to be provided with more concrete reasons, clarifying why the authors think the upscaling should not change the mechanism of rupture.

Dehydration-driven stress transfer triggers intermediate-depth earthquakes

Point-by-point response to reviewers #2

- Answers are in bold type.
- Red color means that text has been modified or added.
- Blue color is used for text that has been moved from the *Results* to the *Discussion*, as well as for paragraphs that have been reorganized (lines 110-123).

Reviewer #1 (Remarks to the Author):

I am happy with the changes the authors made to address my (and others) comments. I recommend that the paper be published.

Reviewer #2 (Remarks to the Author):

I read through the revised paper and the comments from the authors. The revised manuscript looks better than the original one, while I have a few comments relating to my first review.

1. Still the authors haven't answered my first question. The authors commented "local destabilization of ANY HYDROUS PHASE around 600-800°C may lead to a stress transfer triggering mechanical instabilities in a stronger material.", but obviously the destabilization of antigorite at 1.1 GPa doesn't induce any AE activity; only the destabilization of Tlc-like phase does in this condition. The authors' idea, DDST should be applicable for both reactions if it is true. The critical characteristic length either for l_c or ξ might be different in each reaction, but the authors didn't mention it. While temperature measurement has a huge uncertainty, in-situ XRD measurements strongly suggest this issue.

Indeed our first answer was not complete and we apologize. Actually, we argue our mechanism works for any hydrous phase but only if the stronger material around is connected and able to store a sufficient amount of elastic energy. The possibility of generating a mechanical instability will depend on 1) the elastic properties of the surrounding rock, 2) the properties of the dehydrating phase, and 3) on the kinetics of dehydration, which include geometrical factors.

At 1.1 GPa, the ΔV of the reaction is positive ($\approx + 5 \%$, Gasc *et al.*, 2011), which tends to prevent tensile stress accumulations in the olivine skeleton; at 3.5 GPa, the ΔV is negative ($\approx - 1 \%$), which promotes tensile stress accumulation at cluster tips and thus mechanical instabilities. Furthermore, this negative ΔV could promote fluid circulation, which may provide a positive feedback on the reaction, while at 1.1 GPa fluids mobility is smaller, which helps to keep antigorite clusters stable. We have added these two sentences to the discussion (lines 258-262).

A sentence has been added to the manuscript which we hope helps to answer this very good point raised by the reviewer (lines 273-275): “The resulting stress transfer to the surrounding olivine matrix induces a mechanical instability in the aggregates. The stress transfer is more critical when $\Delta V < 0$ because it promotes tensile stress concentrations at dehydrating antigorite cluster tips.”

In addition, at 1.1 GPa, the acoustic activity begins once the antigorite breakdown is overstepped, thus AEs probably have nothing to do with the destabilization of talc-like phase: “Antigorite is observed up to ≈ 920 K” (line 253).

2. The author replied to my second comment in my first review:

“Experiments have been performed on olivine aggregates for decades, investigating a large spectrum of temperature, pressure and strain rate, including by two of the authors of this ms. No brittle behavior within pure polycrystalline olivine was reported/observed in the conditions we used. Hence, in Fig. 3 (prev. Fig. 2), we consider the comparison between the experiments with 0 and 5 vol.% antigorite at 1.1 GPa as an evidence of antigorite being required for brittle behavior.”, but actually this manuscript reported the brittle behavior of the olivine aggregates at 3.5 GPa, it is very strange. Cumulative No. of AE of 100% olivine at 3.5 GPa is 30 to 50% less than those of antigorite/olivine mixtures (just 10 counts difference). Is this ok to say that it is the significant difference (line 165)?

As explained in the manuscript (lines 171-187), the brittle behavior reported in olivine aggregates at 3.5 GPa seems to arise from a completely different mechanism, i.e. the formation of large cataclastic shear bands. As pointed out, the formation of these is rarely associated “with a single dynamic (rapid) shear failure, because the elastic energy is converted into surface energy and dissipated through fractures in the shear zone rather than radiated at high speed.”

We are not arguing that the number of AEs detected is significantly different (although less were detected) but that were detected at much higher stress level, which are not compatible with expected stress in the downgoing cold subducting lithosphere. In addition, our synthetic samples were sintered at relatively low P conditions (1.5 GPa) so that it might be that there is some residual porosity causing local stress concentrations when brought to 3.5 GPa. Our observations of micro-pseudotachylytes on the other hand is an indisputable proof of dynamic shear failures that propagated at peak HP-HT conditions in our dehydrating samples.

A few sentences were added in the manuscript (lines 91-95): “The number of AEs detected in this pure olivine sample is not significantly different (although lower) than in the case of the antigorite-bearing samples. However, AEs were detected at a much higher stress level than for the antigorite-bearing samples and actually at stress levels higher than expected to occur in a cold subducting lithosphere. In addition, our synthetic samples were sintered at relatively low P conditions (1.5 GPa) so that stress concentrations may exist and play a role in the AE triggering at 3.5 GPa.”

Minor comment:

I should have said this in the first review, but in figure 2, it is better for this manuscript to plot the PT path of the Moho or both the Moho and the slab surface rather the PT of the slab surface only. The PT path of the Moho seems to be more important for this research.

This is a good point. However, mantle earthquakes of the LWBP are much deeper than the Moho, especially in cold subduction zones. We would prefer keeping the figure as such, in order to avoid making it heavier.

Reviewer #3 (Remarks to the Author):

Comments on the revised manuscript, “Dehydration-driven stress transfer triggers intermediate-depth earthquakes “ submitted to Nature communications.

This is the second review of the manuscript after a revision responding to the comments made by three reviewers. As I have expressed in my first review, I was impressed with the data presented in the paper including variety of analytical method applied to experimental samples synthesized under high-pressure and high-temperature conditions. With such effort, the manuscript presents a new model explaining the cause of the intermediate-depth, lower Wadati-Benioff plane (LBP) earthquakes. I certainly appreciate the discussion and analytical effort invested in the paper. However, I was less impressed with the presentation of the subject. For the comments on the second review, I primarily focuses my critiques on the presentation of the paper, as well as comments on the reply.

First of all, I had pleasant experience reading the second manuscript. I acknowledge the writing effort and that the increased number of SEM-TEM images (Figure 5, 6, and 7) helped me follow the argument made by the authors. For example, I appreciate the new Figure 6 which displays two identical SEM images with and without a structural interpretation. Overall, the manuscript increased its length from eight to twenty pages main text. I believe the revised, longer, version is more accessible to a wider audience. I have not much comment to add with respect to scientific aspects of the paper.

I have made some comments on which mainly related to structure of the presentation and clarity/precision.

From line 204 to 234, the author deviate from result presentation under the heading of “microstructural observations,” and start discussions related to the significance of microstructure observations. Perhaps, these paragraph should be moved into the discussion section.

Indeed - These two paragraphs (in blue) are now on top of the discussion.

I disagree with how some references are quoted, and the passage should be revised. line 34 - 38, here the reference 10 and 11 (Ranero et al. 2003 and Shillington et al. 2015) are cited to say, “documenting that the lithospheric mantle was partially hydrated tens of kilometers below the Moho”. Ranger et al. 2003 shows the presence of faults with the seismic images covering approximately to 8 km below the MOHO. Then they argue for a possibility of extending such fault to 35 km depth based on an estimate of the brittle-ductile transition of the lithosphere.

We thank the reviewer for this very good remark. We have corrected it to : “[...], recent seismic surveys revealed deep reflections interpreted as bending-related faulting and mantle serpentinization at the Middle America trench (Ranero et al., 2003) and offshore Alaska (Shillington et al., 2015). These observations document that the lithospheric mantle was partially hydrated 8 and 15 km below the Moho respectively, through serpentinization of deep faults, which can be extended to 35 km depth based on an estimate of the brittle-ductile transition of the lithosphere (Ranero et al., 2003)” (lines 34-39).

According to Shillington et al. 2016, their seismic reflection data shows extensive hydration in the uppermost 3-4km of the mantle, not “tens” of kilometer. In addition to such observation, they argue that the correlation between the occurrence of LBP earthquakes and nature of brittle deformation at outer rise demonstrates the presence of hydrated (antigorite) layer, assuming that the intermediate earthquake is caused by antigorite.

This is correct. Shillington et al. 2015 show strong hydration 3-4 km below the Moho, but also little hydration deeper, to an extent of about 15 km (Figure 2 in Shillington et al., 2015). Our results show that extensive dehydration of the mantle is not required. Moreover, the resolution of seismic refraction does not allow measuring 1 to 5 % serpentinization.

For these reasons above, the citation of these papers as the demonstration of the presence of antigorite at the depth only provides an incomplete and circular argument, and furthermore misleading. As far as I am aware, there are no direct observation documenting the presence of serpentine at the 30 km depth below the MOHO. While I would not argue that there are no serpentine at the depth, but I would request the authors to present a leveled overview of the issue.

We thank the reviewer for this demand.

line 97-98, “can be stoichiometrically written as” The exact stoichiometry of antigorite is not what is written here. It is instructive to have an approximate chemical formula but please state that it is an approximate reaction stoichiometry. Antigorite stoichiometry is $M_{(3m-3)}T_{(2m)}O_{(5m)}(OH)_{(4m-6)}$ where $m= 14$ to 23 .

The reviewer is perfectly right and this is differently presented in the manuscript: “The exact antigorite chemical formula is is $M_{(3m-3)}T_{(2m)}O_{(5m)}(OH)_{(4m-6)}$ where $m = 14$ to 23 , M and T are octahedral and tetrahedral sites. Here we use the approximate formula $Mg_3Si_2O_5(OH)_4$. The complete antigorite (atg) dehydration reaction into olivine (ol), enstatite (en) and water, with approximate stoichiometry, is: [...]” (lines 101-104).

Line 193 : The issue raised by Reviewer 2, which is regarding a lack of an exact coincidence of the experimental result and the previous kinetic studies, may not significantly challenge the validity of the result presented here. It should be noted that the in situ kinetic experiments of Perrillat et al., 2005 and Chollet et al., 2011 have an issue of the temperature **accuracy** because their T calibration relies on the equations of state of internal markers. The offset pointed out by Reviewer 2 is well within the expected uncertainty of accuracy. Furthermore, I certainly agree with the authors that such offset can also be explained by the kinetic overstepping caused by heating strategy.

We thank the reviewer for this clear explanation.

Line 153-155 : “Nevertheless, the three orders of magnitude covered by source sizes (Fig. 4a) suggest that our observations could be upscaled”. It is certainly a great achievement to show that the same physical phenomenon can be scaled up to 3 orders of magnitude. However, that alone would not be a sufficient reason to suggest that the same is true for larger scale seismic event in nature. Here, I would like to be provided with more concrete reasons, clarifying why the authors think the upscaling should not change the mechanism of rupture.

Scaling-laws between laboratory experiments and nature are always hard to assess, so that the reviewer’s comment can generally be referred to as the experimentalist’s recurring nightmare.

As you know, it is not possible to deform a 1-km³ piece of peridotite at mantle conditions, at a strain rate of 10⁻¹² s⁻¹. So our only hope is to rely on basic scaling laws, and point out when it seems that the physics and micromechanics of the phenomenon we observe at the lab scale could possibly be the same operating at the field one. Here, we provide two arguments as to why our laboratory observations are probably up-scalable:

- 1) The maximum AE magnitude size in our sample corresponds closely to the size of our specimen, so that we could argue that, with larger specimens, we would get larger AEs. It is reasonable to make this claim because we have observed the G-R scaling in our AE magnitude. A sentence was added to the manuscript (lines 157-164): “The maximum AE magnitude size in our sample corresponds to the size of our specimen (next section: *Microstructural observations*). The minimum value of the AE magnitude we detected probably corresponds to our detection limit. The Gutenberg-Richter scaling thus suggests that larger AE magnitudes would have been observed if our sample were larger. It has been recently shown that the fracture energy dissipated during an earthquake rupture scales with the size of the seismic asperities (*Passelègue et al., 2016*), with an apparent continuous scaling law from the laboratory scale, to the field scale. Thus, our observation of three orders of magnitude of seismic source sizes (Fig. 4a) suggests that our observations can be upscaled.”
- 2) Our model also provides a scaling, because it has been shown that the fracture energy, and thus the nucleation length, scales with the size of the seismic asperities (*Ohnaka, 2003 ; Passelègue et al., 2016*). Furthermore, cluster-lengths scales, which here depend on the grain size and the olivine-antigorite proportions, can easily be scaled to the field scale, where they could correspond to unaltered peridotite bodies, with volumes of the order of 1 to 100 km³, bounded by discrete fault zones at various degrees of serpentinization, which may constitute the equivalent of our olivine sub-volumes. The text was modified accordingly (lines 338-342): “In addition, our model provides a possible scaling of our laboratory observations to the field, since it relies on three up-scalable length scales. Fracture energy, and thus the nucleation length, scales with the size of seismic asperities (*Passelègue et al., 2016*). In the laboratory, olivine- and antigorite-cluster-lengths scales depend on the grain size and the olivine-antigorite proportions.”
- Finally, the experimentally deformed samples contain laboratory equivalents to pseudotachylytes. These features have been observed in different places in nature, especially in outcrops of ancient deep crust or upper mantle. For instance, pseudotachylytes are observed in peridotite bodies (e.g. Obata et al., 1995; Ueda, 2008; Souquière & Fabbri, 2010, Deseta et al., 2014a 2014b), thus their generation is expected at mantle depth. Thus

the laboratory pseudotachylytes strongly suggest that a similar mechanism is at play at laboratory scale and field scale.

For these reasons we believe that our experiments have relevance to what is happening at the field scale.

Reviewers' Comments:

Reviewer #2 (Remarks to the Author)

I read through the revised paper and the "rebuttal", and feel that it is ready for publication. I hope this manuscript will be published very soon.